# Accelerate Vertical Federated Adversarial Learning with Dual-level Decoupled Backpropagation

## Abstract

Vertical Federated Learning (VFL) involves multiple participants collaborating to train models on distinct feature sets from the same data samples. The distributed deployment of VFL models renders them vulnerable to adversarial perturbations during inference, motivating the need to visit the VFL robustness problem. Adversarial Training (AT) is the predominant approach for enhancing model robustness. However, its application in VFL, termed Vertical Federated Adversarial Learning (VFAL), faces significant computational challenges: Generating adversarial examples in AT requires *iterative full propagations across participants with heavy computation overload*, resulting in VFAL training time far exceeding those of regular VFLs. To address this challenge, we propose ***DecVFAL***, an accelerated **VFAL** framework through a novel **Dec**oupled backpropagation incorporating a *dual-level decoupled mechanism to enable lazy sequential and decoupled parallel backpropagation*. Lazy sequential backpropagation sequentially updates the adversarial example using timely partial derivatives with respect to the bottom module and delayed partial derivatives for the remaining modules. Decoupled parallel backpropagation updates these delayed partial derivatives by utilizing module-wise delayed gradients, enabling asynchronous parallel backpropagation with flexible partitions that align with VFL's distributed deployment. Rigorous theoretical analysis demonstrates that despite introducing multi-source approximate gradients due to the dual decoupled mechanism and the techniques from the existing VFL methods, *DecVFAL* achieves a $\mathcal{O}(1/\sqrt{\mathcal{K}})$ convergence rate after $\mathcal{K}$ iterations, on par with regular VFL systems. Experimental results show that, compared to existing methods, *DecVFAL* ensures competitive robustness while significantly achieving about $3 \sim 10$ times speed up on various datasets.

## 1 Introduction

Federated learning (FL) enables collaborative training of deep learning models among distributed participants without sharing raw data McMahan et al. (2016). Conventionally, most FL research considers Horizontal Federated Learning (HFL), which assumes distributed clients possess data with identical features but varying sample spaces Zhao et al. (2021). In contrast, *Vertical Federated Learning (VFL)* assumes distributed clients share the same samples but have different features Liu et al. (2024); Wei et al. (2022). VFL model comprises a server-maintained top model and client-side bottom models that map local data features to embeddings. During inference, each client computes the local embedding of data features and uploads to the server through a communication channel for prediction Liu et al. (2024). Due to its advantages in facilitating data collaboration across multiple industries, VFL has gained increasing attention in various domains such as recommendation systems Cui et al. (2021); Yuan et al. (2022), finance Long et al. (2020); Chen et al. (2021a), healthcare Song et al. (2021); Cha et al. (2021), and emerging applications Teimoori et al. (2022); Ge et al. (2022).

Machine Learning (ML) models have demonstrated vulnerability to *adversarial attacks*, carefully crafted inputs designed to induce misclassification during inference. Recent studies highlight that this susceptibility becomes even more pronounced in the VFL context due to its decentralized architecture Huang et al. (2024); Duanyi et al. (2023). Adversarial attacks in VFL can manifest in multiple forms: through malicious or colluding clients perturbing local features of raw data Pang et al. (2022); Qiu

et al. (2022), or via third-party adversary intercepting and altering embeddings during client-server communication Duanyi et al. (2023). These diverse attacks underscore *the unique security challenges in VFL systems, motivating the urgent need to address the VFL robustness problem.*

Extensive research has been conducted on defenses against adversarial attacks, with *Adversarial Training (AT)* emerging as the most empirically robust approach to date Tramèr et al. (2018). AT is a min-max robust training method that minimizes the worst-case training loss at adversarially perturbed examples Madry et al. (2017). The deployment of AT in the FL paradigm, termed Federated Adversarial Learning (FAL), has garnered attention, with a particular focus on HFL scenarios, where each participant maintains a complete copy of the model Li et al. (2023). These studies incorporate AT into clients' local training steps and focus on non-IID settings and secure aggregation solutions Li et al. (2023); Deng et al. (2020); Bhagoji et al. (2019); Zizzo et al. (2020); Zhang et al. (2022a). However, in VFL scenarios, a single global model is partitioned and distributed among the server and clients, resulting in a different architecture for Vertical Federated Adversarial Learning (VFAL). To the best of our knowledge, *VFAL has yet to be thoroughly investigated in the current literature.*

*Due to layer-wise distributed deployment, VFAL presents unique computational efficiency challenges.* Adversarial sample generation during AT is computationally intensive, requiring sequential forward and backward propagation to calculate gradients with respect to the input for iterative refinement Madry et al. (2017). In VFL context, inherent sequential dependencies across layers cause participants' models to remain idle until receiving necessary information (embeddings or gradients) from adjacent layers on other participants (Figure 1-left). Consequently, the training time for VFAL significantly exceeds that of regular VFLs. To illustrate, VFAL using PGD-20 requires about 20 times more computational cost than regular VFL due to 20 iterations needed to generate each adversarial example.

Several works have focused on accelerating AT-based robust training, but they are designed for centralized model training without consideration for adaptation to VFAL. Examples include YOPO estimates the gradient on the input by only propagating the first layer Zhang et al. (2019), FreeAT reuses gradients for multiple steps to update both adversarial examples and model parameters, Shafahi et al. (2019), Amata adjusts the number of inner maximization steps with an annealing mechanism Ye et al. (2021), Bhat & Tsipras (2019) propose asynchronously generating adversarial examples leveraging data parallelism, and FGSM-PGK assembles the prior-guided initialization and model weights Jia et al. (2024). Another line of research explores the design of computational efficient vanilla VFL frameworks, including multiple client updates Zhang et al. (2022b), asynchronous coordination Li et al. (2020), compression Castiglia et al. (2022); Li et al. (2020), sample and feature selection Castiglia et al. (2023); Huang et al. (2022) one-shot communication Wu et al. (2022); Cha et al. (2021). While these studies have made significant strides in improving the computational efficiency of VFL, they lack a comprehensive investigation into the integration with VFAL framework. Taking into account these observations and challenges, a natural question arises:

*In light of the intensive adversarial sample generation and inherent sequential dependencies, how can we accelerate VFAL training while maintaining robust performance?*

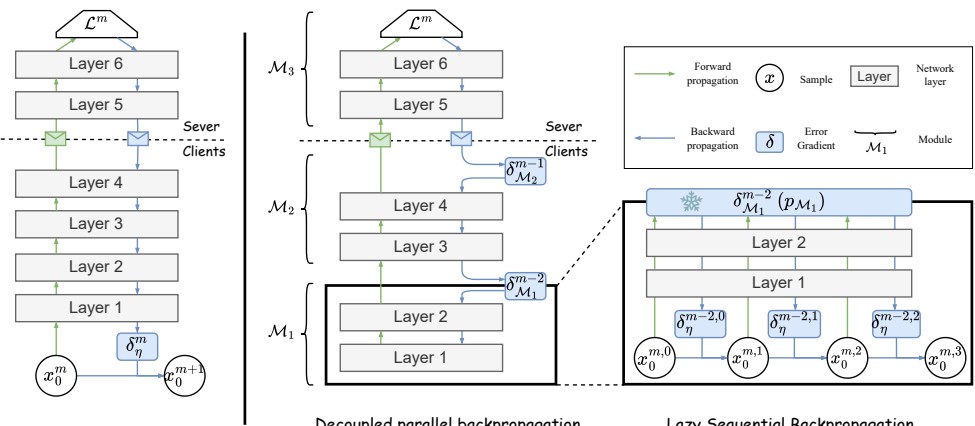

Figure 1: Comparison of one-time full propagation for adversarial example generation: VFL with PGD (left) versus DecVFAL (right).

To tackle the computational efficiency challenge in training robust VFL models, we propose *DecV-FAL*, an accelerated **VFAL** framework through a novel **Dec**oupled backpropagation incorporating a *dual-level decoupled mechanism* (Figure 1-right). DecVFAL first decouples the bottom module from the remaining modules and introduces *lazy sequential backpropagation*, which periodically treats the partial derivatives of the remaining modules as fixed and utilizes timely partial derivatives for the bottom module to execute multiple sample updates sequentially, avoiding frequent complete gradient propagation. Furthermore, while updating the adversarial samples at the bottom module, DecVFAL updates the partial derivatives of the remaining modules through *decoupled parallel backpropagation*, where each module independently updates its partial derivatives with module-wise delayed gradients on separate processors, achieving asynchronous parallel backpropagation.

**Contributions (i)** We propose DecVFAL, which incorporates a dual-level decoupled mechanism to enable lazy sequential and decoupled parallel backpropagation, significantly accelerating VFAL training while maintaining robust performance. **(ii)** Our rigorous theoretical analysis reveals that despite the introduction of multi-source approximate gradients, DecVFAL maintains an $\mathcal{O}(1/\sqrt{\mathcal{K}})$ convergence rate after $\mathcal{K}$ iterations, matching that of standard VFLs, underscoring the superiority of DecVFAL. **(iii)** Comprehensive experimental evaluations demonstrate that DecVFAL not only achieves competitive robust performance but also delivers a remarkable $3 \sim 10$ fold acceleration compared to existing adversarial training methods compatible with VFL.

## 2 RELATED WORKS

**Adversarial Attack in VFL.** Research highlights the need for robust VFL models Ye et al. (2024), while introducing novel adversarial attack techniques Duanyi et al. (2023); Chen et al. (2022). In relaxed VFL protocols, where clients can access the server model and outputs from other clients Luo et al. (2021); Wang (2019); Lundberg & Lee (2017), a wide range of white-box adversarial attacks Madry et al. (2017); Carlini & Wagner (2017); Croce & Hein (2020); Kurakin et al. (2016) become feasible through *malicious and colluding clients*. Standard VFL protocols, despite restricting critical information, remain vulnerable to black-box adversarial attacks Chen et al. (2017). Additionally, Chen et al. (2022) employs a GAN-based method with a surrogate model and semi-supervised learning to generate performance-impairing perturbations. Further expanding the threat landscape, Duanyi et al. (2023) explores third-party adversaries through an online optimization method that disrupts inference, integrating adversarial example generation with corruption pattern selection.

**Adversarial Training.** AT enhances model robustness by incorporating adversarial examples, with its effectiveness depending on the strength of those examples Goodfellow et al. (2014). While non-iterative attacks like FGSM offer some resilience, they remain vulnerable to more advanced methods Kurakin et al. (2016). Projected Gradient Descent (PGD) Madry et al. (2017) provides superior robustness against obfuscated gradient defenses Athalye et al. (2018) but is computationally expensive due to frequent adversarial updates. FreeAT Shafahi et al. (2019) combines the updates of adversarial examples and model parameters in one backward pass, YOPO Zhang et al. (2019) focuses on adversarial example updates at first-layer, and FreeLB Zhu et al. (2019) accumulates gradients and update parameters after completing adversarial iterations. While these methods offer promising approaches to balance robustness and efficiency in AT, their applicability and effectiveness within the VFAL framework remain unexplored, highlighting a critical gap in current research.

**Decouple Training.** The inherently sequential nature of forward and backward propagation in neural network training has long been a focus of optimization, with researchers proposing various innovative methods to decouple the process and improve computational efficiency. Notable contributions include the Alternating Direction Method of Multipliers (ADMM), which decomposes the optimization problem into smaller, more manageable subproblems, facilitating parallel processing Taylor et al. (2016). Synthetic Gradients enable asynchronous updates by predicting gradients for each layer, reducing dependencies between network components Jaderberg et al. (2017). The delayed Gradient Method allows for parallel processing of different network sections, potentially speeding up training, by introducing a temporal shift in gradient computation Huo et al. (2018b;a); Zhao et al. (2024). Lifted Machines involves transforming the network architecture to create opportunities for parallelization, thereby improving computational efficiency Gu et al. (2020); Li et al. (2019).

## 3 PROBLEM DEFINITION

**Notations.** In the VFL framework consisting of one server and $C$ clients Liu et al. (2024), we consider a classifier represented by a $T$-layer deep neural network $f(\Theta; x)$, where $x$ denotes the input and $\Theta$ the set of trainable parameters. The training dataset is denoted as $\{(x_{0,i}, y_i)\}_{i=1}^{\mathcal{S}}$, with $\mathcal{S}$ representing the total number of samples. Each sample is composed of features from different clients, specifically $x_{0,i} = [x_{0,i,(1)}, \ldots, x_{0,i,(C)}]$. The classifier comprises client models $[f_{(1)}, \ldots, f_{(C)}]$ parameterized by $[\theta_{(1)}, \ldots, \theta_{(C)}]$ and a server model $f_s$ parameterized by $\psi$. The classifier function is expressed as $f(\Theta, x_{0,i}) = f_s\{\psi; f_{(1)}[\theta_{(1)}; x_{0,i,(1)}], \ldots, f_{(C)}[\theta_{(C)}; x_{0,i,(C)}]\}$, where $\Theta = [\theta_{(1)}, \ldots, \theta_{(C)}, \psi]$. All notations used in this paper are summarized in Appendix B.1.

**Vertical Federated Adversarial Learning.** Building upon the standard VFL models and the minimax problem in AT, a $T$-layer neural network $f$ is defined recursively as: $x_t = f_t(x_{t-1}, \Theta_t), t = 1, \ldots, T$, where $x_t$ are the output of the $t$-th layer, $\Theta_t$ are the parameters of layer $f_t$, $\Theta$ denotes the concatenation of $(\Theta_t)_{1 \leq t \leq T}$. VFAL addresses problems of the following general form:

$$
\begin{aligned}
\min_{\Theta} \max_{\|\eta_i\|_\infty \leq \epsilon} \quad & \sum_{i=1}^{\mathcal{S}} \mathcal{L}(x_{T,i}; y_i) + \sum_{i=1}^{\mathcal{S}} \sum_{t=1}^{T} R_t(\Theta_t; x_{t-1,i}) \\
\text{subject to} \quad & x_{t,i} = f_t(\Theta_t; x_{t-1,i}), \quad i = 1, \ldots, \mathcal{S}, \quad t = 2, \ldots, T \\
& x_{1,i} = f_1(\Theta_1; x_{0,i} + \eta_i), \quad i = 1, \ldots, \mathcal{S}
\end{aligned}
\tag{3.1}
$$

where $t_c$ is the number of client model layers, for $t_c < t \leq T$, $\Theta_t = \psi_{t-t_c}$ are the server model parameters; for $0 < t \leq t_c$, $\Theta_t = [\theta_{t,(1)}, \ldots, \theta_{t,(C)}]$ are the client model parameters. $\eta_i = \eta_{i,(1)}, \ldots, \eta_{i,(C)}$ represents adversarial perturbations on sample $i$, constrained by $\|\eta\|_\infty \leq \epsilon$ (a non-negative scalar $\epsilon$ limits the perturbation magnitude). $\mathcal{L}(\cdot; y)$ is the loss function, and $x_{T,i} = f(\Theta; x_{0,i} + \eta_i)$ is the final output: $x_{T,i} = f(\Theta; x_{0,i} + \eta_i) = f_T(\Theta_T; f_{T-1}(\Theta_{T-1}; \ldots f_1(\Theta_1; x_{0,i} + \eta_i) \ldots))$, $R_t$ is a potential regularization term for layer $f_t$.

## 4 METHODOLOGY

### 4.1 REVISIT BACKPROPAGATION FOR VFAL TRAINING

Addressing the problem (3.1), VFAL training involves two types of backpropagation. The primary computational cost of VFAL arises from the multi-step gradient ascent, therefore, this paper focuses on the acceleration of the adversarial perturbation backpropagation.

**Adversarial Perturbation Backpropagation.** For inner maximization, we keep the model parameter fixed. The adversarial perturbations are updated via multi-step gradient ascent: $\eta^{\ell+1} = \eta^\ell + \alpha_\eta \nabla_\eta \mathcal{L}(\eta^\ell)$, where $\mathcal{L}(\eta^\ell) = \mathcal{L}(f(\Theta^k; x_0 + \eta^\ell); y)$, $\ell$ is the inner iteration index, $k$ is the outer iteration index and $\alpha_\eta$ is the step size. In the forward pass, the activations of all layers are calculated from $t = 1$ to $T$. In the backward pass, chain rule is applied to compute these gradients and propagate the error gradients through the network from $t = T$ to 1: $\frac{\partial \mathcal{L}(\eta^\ell)}{\partial x_{t-1}^\ell} = \frac{\partial x_t^\ell}{\partial x_{t-1}^\ell} \frac{\partial \mathcal{L}(\eta^\ell)}{\partial x_t^\ell}$. The computation at layer $t$ is dependent on the error gradient $\frac{\partial \mathcal{L}(\eta^\ell)}{\partial x_t^\ell}$ from layer $t + 1$. The gradient to $\eta$ is calculated at first layer: $\nabla_\eta \mathcal{L}(\eta^\ell) = \frac{\partial \mathcal{L}(\eta^\ell)}{\partial \eta^\ell} = \frac{\partial x_1^\ell}{\partial \eta^\ell} \cdot \frac{\partial \mathcal{L}(\eta^\ell)}{\partial x_1^\ell}$.

**Model Parameter Backpropagation.** After obtaining the perturbation $\eta$ through inner maximization, we update $\Theta$ via gradient descent using $\nabla_{\Theta_t} \mathcal{L}(\Theta^k) = \frac{\partial x_t^k}{\partial \Theta_t^k} \frac{\partial \mathcal{L}(\Theta^k)}{\partial x_t^k}$ computed during backpropagation w.r.t. the parameters $\Theta$.

**Backward Locking.** Consistent with VFAL's distributed deployment, we can partition a $T$-layer neural network into $\mathcal{M}_K \ll T$ modules. The above formulation reveals that the partial derivatives computation in module $f_{\mathcal{M}_k}$ remains dependent on the error gradient from module $f_{\mathcal{M}_{k+1}}$. This creates a "lock" that prevents layers/modules from partial derivative updating until they receive backward results from their dependent counterparts. As shown in Figure 1-left, each adversarial example update of PGD in VFL context requires sequential propagating error gradients from the output layer back to the input layer.

## 4.2 Dual-Level Decoupled Mechanism

To address the training efficiency bottleneck, DecVFAL introduces a dual-level decoupled mechanism that utilizes module-wise staleness to untether the dependencies across layers inherent in VFAL. As shown in Figure 1-right, DecVFAL utilizes delayed gradients to eliminate backward locking, enabling module-wise asynchronous backpropagation. It restricts perturbation update propagations to the bottom model to reduce full propagations and utilizes gradients from disparate iterations to achieve parallel backward computation. We summarize the proposed algorithm in Algorithm 1 and present the details of DecVFAL in the following sections.

---

**Algorithm 1:** DecVFAL

**Input:** Learning rates $\alpha_\eta, \alpha_\psi, \alpha_\theta$; Train set $\{X, Y\}$.
**Output:** Model parameters $\Theta = \{\theta_{(1)}, \theta_{(2)}, \ldots, \theta_{(C)}, \psi\}$.

1 **Initialization:** Clients and Server initialize model parameters $\theta_{(1)}, \theta_{(2)}, \ldots, \theta_{(C)}, \psi$;
2 **while** *not convergent* **do**
3    Randomly select a sample $x$;
4    **for** $m = 1$ *to* $M$ **do**
5       $\mathcal{L}^m \leftarrow f(x_0 + \eta^{m,n})$;
6       **for** *k = 1 to* $\mathcal{M}_K$ **in parallel do**
7          **if** *k = 1* **then**
8             **for** *n = 0 to N-1* **do**
9                $x_{\mathcal{M}_1}^{m,n} \leftarrow f_{\mathcal{M}_1}(x_0 + \eta^{m,n})$;
10                Updates adversarial perturbation:
11                   $\eta^{m,n+1} \leftarrow \eta^{m,n} + \alpha_\eta p_{\mathcal{M}_1} \nabla_\eta f_{\mathcal{M}_1}$;
12          Backward computation with delayed gradient $\frac{\delta \mathcal{L}^{m-\tau_k}}{\delta x_{\mathcal{M}_k}}$:
13            $\frac{\delta \mathcal{L}^{m-\tau_k}}{\delta x_{t-1}} \leftarrow \frac{\delta x_t}{\delta x_{t-1}} \frac{\delta \mathcal{L}^{m-\tau_k}}{\delta x_t}, t \in (t_{\mathcal{M}_{k-1}}, t_{\mathcal{M}_k}]$;
14       **for** *each client c* **do**
15          Update client model parameters $\theta_{(c)}^{k+1} \leftarrow \theta_{(c)}^k - \alpha_\theta \nabla_\theta \mathcal{L}(f(x_0 + \eta^{m,n}))$ ;
16       Update server model parameters $\psi^{k+1} \leftarrow \psi^k - \alpha_\psi \nabla_\psi \mathcal{L}(f(x_0 + \eta^{m,n}))$.

---

**Lazy Sequential Backpropagation.** A key observation in VFAL is that the adversarial perturbation is directly coupled with the bottom module of the network. This insight allows us to decouple the bottom module $f_{\mathcal{M}_1}(\Theta_{\mathcal{M}_1}; x_{\mathcal{M}_1})$ and the remaining modules $f_{\tilde{\mathcal{M}}_1}(\Theta_{\tilde{\mathcal{M}}_1}; x_{\mathcal{M}_1})$, where $f_{\tilde{\mathcal{M}}_1} = f_{\mathcal{M}_2} \circ f_{\mathcal{M}_3} \circ \ldots f_{\mathcal{M}_K}$, and $x_{\mathcal{M}_1}$ is the output of bottom module. The VFAL classifier can be rewritten as: $f(\Theta; x_0 + \eta) = f_{\tilde{\mathcal{M}}_1}(\Theta_{\tilde{\mathcal{M}}_1}; f_{\mathcal{M}_1}(\Theta_{\mathcal{M}_1}, x_0 + \eta))$. PGD-based AT (PGD-r) involves $r$ sweeps of forward and backward propagation to generate an adversarial example, resulting in extensive computational cost. To mitigate this, we introduce a "lazy" backpropagation mechanism by freezing a slack variable $p_{\mathcal{M}_1}$.

$$p_{\mathcal{M}_1} = \nabla_{f_{\tilde{\mathcal{M}}_1}} \left( \mathcal{L}(f_{\tilde{\mathcal{M}}_1}(f_{\mathcal{M}_1}(\Theta_{\mathcal{M}_1}; x_0 + \eta)), y) \right) \cdot \nabla_{f_{\mathcal{M}_1}} \left( f_{\tilde{\Theta}_{\mathcal{M}_1}}(f_{\mathcal{M}_1}(\Theta_{\mathcal{M}_1}; x_0 + \eta)) \right) \quad (4.1)$$

$p_{\mathcal{M}_1}$ is obtained after each full backpropagation. The adversarial perturbation $\eta$ is updated using $p_{\mathcal{M}_1}$ and $N$-step gradient ascent, while keeping the network parameters $\Theta$ fixed (lines 7-11 in Algorithm 1). As shown in Figure 2, DecVFAL accesses the data $M \times N$ times for each adversarial example generation while only requiring $M$ full forward and backward propagation, where $M \ll r$.

This frozen slack variable introduces an oracle error in adversary updating, resulting in a delayed gradient. Inspired by the optimal control theory Li et al. (2018); Li & Hao (2018); Seidman et al. (2020) and under **Assumptions** in (B.2), we bound costate difference at bottom module in Lemma 1.

**Lemma 1.** ***Bound the costate difference at the bottom module.*** *There exists a constant $G'$ dependent on $T$ and $K$ such that for all $n \in \{0, \ldots, N\}$, $m \in \{0, \ldots, M\}$, and $i \in \{1, \ldots, S\}$:*

$$\left\| p_{\mathcal{M}_1,i}^{m-\tau_1,0} - p_{\mathcal{M}_1,i}^{m,N} \right\| \leq G' \alpha_\eta \left( \mathcal{M}_K N - 1 \right). \quad (4.2)$$

*Where $G' = TK^{T+1}(K^T + T(T-1)K^{2T-2} + TK^{2T})$, $m$ is the iteration index of full propagation, $\tau_1$ is the delay of module $\mathcal{M}_1$ raised from parallel backpropagation.*

**Decoupled Parallel Backpropagation.** We decouple backpropagation across the entire network using delayed gradients, enabling parallel updates of the partial derivatives in the remaining modules for lazy sequential backpropagation. The forward pass is performed from module 1 to module $\mathcal{M}_K$. In backward pass, all modules except the last one store delayed error gradients, allowing to perform the backward computation without locking. The module $f_{\mathcal{M}_k}$ keeps the stale error gradient $\frac{\delta \mathcal{L}^{m-\tau_k}}{\delta x_{\mathcal{M}_k}}$, $\tau_k = \mathcal{M}_K - \mathcal{M}_k$. Therefore, aside from the bottom module performing lazy backpropagation, the backward computation in the remaining modules $f_{\mathcal{M}_k}$ is as follows:

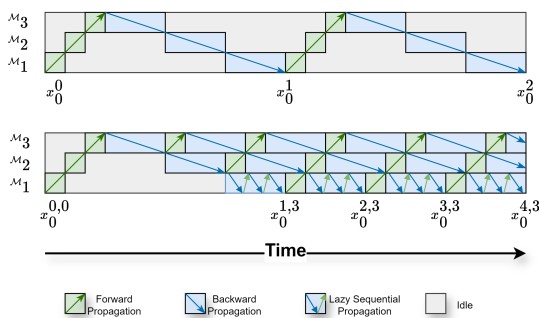

Figure 2: Comparison of computation time: VFL with PGD (up) versus DecVFAL (down). DecVFAL updates adversarial examples $4 \times 3$ times in approximately the same time as performing 2 PGD updates.

$$\frac{\delta \mathcal{L}^{m-\tau_k}}{\delta x_{t-1}} = \frac{\delta x_t}{\delta x_{t-1}} \frac{\delta \mathcal{L}^{m-\tau_k}}{\delta x_t}, t \in (t_{\mathcal{M}_{k-1}}, t_{\mathcal{M}_k}] \tag{4.3}$$

Meanwhile, each module also receives a gradient from the dependent module for further computation. The delayed gradients in all modules are of different time delays. From module 1 to module $\mathcal{M}_K$, their corresponding time delays $\tau_k$ are from $\mathcal{M}_K - 1$ to 0. Delay 0 indicates that the gradients are up-to-date. In this way, we break the backward locking and achieve decoupled parallel backpropagation.

To showcase the flexibility of DecVFAL's module partitioning, we implement the proposed framework within a hybrid cascaded VFL architecture Wang et al. (2024). We analyze the errors caused by *multi-source approximate gradients* due to existing VFL and DecVFAL in Lemma 2.

**Lemma 2.** ***Bound the gradient to*** $\eta$. *Under hybrid cascaded VFL architecture, the gradient $\nabla_\eta \mathcal{A}$ respect to $\eta$ involves estimation gradient $\nabla_\eta \hat{\mathcal{A}}$ from Zeroth Order Optimization (Appendix A.5) and compression gradient $\hat{\nabla}_\eta \mathcal{A}$ (Appendix A.6). Under the **Assumption 1**, and **Lemma 3, 5**, at the $i$-th sample and $k$-th iteration, the pseudo-partial derivative for $\eta$ satisfies the following inequality $\hat{\eta}_i = \underset{\substack{m=1,\ldots,M \\ n=1,\ldots,N}}{argmin} \left\| \hat{\nabla}_\eta \hat{\mathcal{A}}_i (\eta_i^{m,n}, \psi_i, \theta_i) \right\|$, we define $G = KG'$, $\alpha_x < \frac{1}{L_{\eta\eta}}$ then:*

$$\mathbb{E} \left\| \hat{\nabla}_\eta \hat{\mathcal{A}}_i(\hat{\eta}_i, \psi_i, \theta_i) \right\|^2 \leq \left[ D(\mathcal{X}) L_{\eta\eta}^2 \left(1 - \frac{z}{L_{\eta\eta}}\right)^{MN+1} + \frac{2G^2}{L_{\eta\eta}} (\mathcal{M}_K N - 1)^2 \left(\frac{2}{z} + \frac{1}{2L_{\eta\eta}}\right) \right]$$
$$\times 3 \left( H_\theta^2 C \mathcal{E}^k + \frac{L^2 \mu^2 d^2}{4} + K^2 \right) \tag{4.4}$$

## 4.3 ACCELERATION OF DECVFAL

DecVFAL uses the dual-level decoupled mechanism to accelerate the VFAL training process. Specifically, lazy sequential backpropagation allows us to update $M * N$ times to generate adversarial samples with only $M$ full propagations. Empirically, DecVFAL achieves comparable results only requiring setting $M * N$ a litter larger than $r$ of PGD-r. Furthermore, assuming that the time for full propagation is $\mathcal{T}$, decoupled parallel backpropagation reduces this approach to $\frac{\mathcal{T}}{\mathcal{M}_K}$. It is worth noting that prior research employs parallelism for model training using delayed gradients, where updates occur after each propagation. This approach precludes parallelization of forward and backward propagation, limiting acceleration to $\mathcal{T}_{for} + \frac{\mathcal{T}_{back}}{\mathcal{M}_K}$ Huo et al. (2018b;a). In contrast, our method achieves acceleration to $\frac{\mathcal{T}}{\mathcal{M}_K}$, since adversarial sample generation maintains constant parameters, enabling concurrent forward and backward propagation. overall, the computation time for DecVFAL to complete an adversarial example generation is $\frac{M*\mathcal{T}}{\mathcal{M}_K}$, much smaller than $r * \mathcal{T}$ of PGD-r.

## 5 CONVERGENCE ANALYSIS

**Assumptions:** The formal definition and detailed discussion of the assumptions are in Appendix B.2. We make several crucial assumptions: the functions $f_t$, $f_c$, $\mathcal{L}$, and $R_t$ are $K$-Lipschitz continuous in $x$, uniformly with respect to $\theta$ and $\psi$, the gradient of the adversarial loss function, $\nabla \mathcal{A}_i(\eta, \psi, \theta)$, satisfies Lipschitz conditions (Assumption 1); the adversarial loss function $\mathcal{A}_i(\eta, \psi, \theta)$ possesses an unbiased gradient (Assumption 2) and is characterized by bounded Hessian matrices $H_\psi$ and $H_\theta$ (Assumption 3), as well as bounded block-coordinate gradients $Q_\psi$ and $Q_\theta$ (Assumption 4); $\mathcal{A}_i(\eta, \psi, \theta)$ exhibits $z$-strong concavity with respect to $\eta$ (Assumption 5).

**Theorem 1.** Under Assumptions (1, 2, 3, 4), if the step sizes satisfy $\alpha_\eta < 1/L_{\eta\eta}$, $\alpha_m = \min\{\alpha_\psi, \alpha_\theta\}$, $\alpha_M = \max\{\alpha_\psi, \alpha_\theta\}$, and $\frac{\alpha_M}{\alpha_m} < \infty$. Also, $\eta_i^* = \arg\max_\eta \mathcal{A}_i(\eta, \psi, \theta)$ and $\Lambda = \mathcal{R}\left(\eta^{*,0}, \psi^0, \theta^0\right) - \inf_k(\mathcal{R}\left(\eta^{*,k}, \psi^k, \theta^k\right))$. Then the following inequality holds:

$$\frac{1}{\mathcal{K}} \sum_{k=0}^{\mathcal{K}-1} \mathbb{E}\left[||\nabla \mathcal{R}\left(\eta^{*,k}, \psi^k, \theta^k\right)||^2\right] \le \mathcal{I}_1 + \mathcal{I}_2 + E_p + E_c + E_z \qquad (5.1)$$

where $\mathcal{I}_1 = \frac{2\Lambda}{\alpha_m \mathcal{K}}$, $\mathcal{I}_2 = \frac{2L_\star \alpha_M^2 \sigma_\psi^2}{\alpha_m} + \frac{2L_\star \alpha_M^2 \sigma_\theta^2}{\alpha_m}$, $E_p = \frac{3\alpha_M K^2 \pi(M,N)}{\alpha_m z^2}\left(2\xi_\psi L_\star^2 + 3\xi_\theta L_\star^2\right)$, $E_c = \mathcal{E}\frac{\alpha_M}{\alpha_m}\left(2\xi_\psi H_\psi^2 C + 3\xi_\theta Q_\theta^2 H_\theta^2 C + \frac{3H_\theta^2 C\pi(M,N)}{z^2}(2\xi_\psi L_\star^2 + 3\xi_\theta L_\star^2)\right)$, $E_z = \mu^2\left(\frac{3\alpha_M \xi_\theta L_\star^2 d^2}{4\alpha_m} + \frac{3\pi(M,N)L_\star^2 d^2 a_M \xi_\psi}{2a_m z^2} + \frac{9\pi(M,N)L_\star^2 d^2 a_M \xi_\theta}{4a_m z^2}\right)$, $\xi_\theta = \{1 + L_\theta \alpha_M\}$, $\xi_\psi = \{1 + L_\psi \alpha_M\}$, and $L_\star = max\{L, L_\psi, L_\theta, L_{\psi\eta}, L_{\theta\eta}\}$, $\mathcal{K}$ is the total number of iterations.

Term $\mathcal{I}_1$ is typical for convergence of first-order optimization algorithms on smooth non-convex functions; Term $\mathcal{I}_2$ is typical for stochastic gradient descent; Term $E_c$ is the errors during forward communication due to compression; Term $E_z$ is the errors due to zeroth-order optimization; Term $E_p$ is errors due to dual-level decoupled backpropagation for adversarial sample generation.

**Corollary 1.** *If we choose $\alpha_\theta$ and $\alpha_\psi$ as $\frac{1}{\sqrt{\mathcal{K}}}$, $\mu = \frac{1}{\mathcal{K}^{\frac{1}{4}}}$, $\mathcal{E} = \mathcal{O}(\frac{1}{\sqrt{\mathcal{K}}})$, $\Gamma = \mathcal{O}(\frac{1}{\sqrt{\mathcal{K}}})$, we can derive the sublinear convergence rate:*

$$\frac{1}{\mathcal{K}} \sum_{k=0}^{\mathcal{K}-1} \mathbb{E}\left[||\nabla \mathcal{R}\left(\eta^{*,k}, \psi^k, \theta^k\right)||^2\right] \le \mathcal{O}(\frac{1}{\sqrt{\mathcal{K}}}) + \mathcal{O}(\frac{N}{M}) \qquad (5.2)$$

By constraining multi-source approximate gradients, we demonstrate the sublinear convergence rate $\mathcal{O}(\frac{1}{\sqrt{\mathcal{K}}})$. The term $\mathcal{O}(\frac{N}{M})$ refers to a similar result in Seidman et al. (2020), revealing the dependence on $M$ and $N$. We have the partial derivative of $\frac{\partial \pi(M,N)}{\partial N}$ to $N$: $D(\mathcal{X})L_{\eta\eta}^2\left(1 - \frac{z}{L_{\eta\eta}}\right)^{MN+1}\ln\left(1 - \frac{z}{L_{\eta\eta}}\right)M + \frac{4G^2\mathcal{M}_K}{L_{\eta\eta}}\left(\frac{2}{z} + \frac{1}{2L_{\eta\eta}}\right)(\mathcal{M}_K N - 1)$. $\pi(M,N)$ decreases concerning $M$, implying that $M$ should be set large as tolerated according to the communication budget. $\pi(M,N)$ is convex in $N$, the second-order derivative of $\pi(M,N)$ concerning $N$ is greater than 0, therefore, the value of $N$ should increase before the partial derivative with respect to $N$ becomes positive. After that, we need to control the value of $N$ not to be too large, otherwise the model obtains a lower robust accuracy. We conducted ablation experiments and verified this dependence of $M$ and $N$ on the MNIST dataset (Section 6.4).

**Proof Sketch.** We begin by transforming the original min-max optimization problem into a Hamiltonian system (Appendix A.4). The convergence analysis leverages three types of approximate gradients: delayed gradient (Lemma 1 and Lemma 2), compression gradient (Lemma 5), and estimated gradient (Lemma 3). We establish the global convergence of the framework by proving that the loss function $\mathcal{L}(\eta, \psi, \theta)$ is L-smooth (Assumption 1). By combining the results from the $M$ loop, $N$ loop, and outer loop analyses, we demonstrate that the model parameters converge asymptotically (Theorem 1). In Appendix B, we provided detailed proof of the convergence analysis of DecVFAL.

## 6 EXPERIMENTS

We conducted a comprehensive series of experiments to evaluate the effectiveness of our proposed DecVFAL framework. As baselines, we implemented several established AT methods applied to the

standard VFL framework, as well as the well-known VFL acceleration mechanisms. Our results show that DecVFAL achieves the optimal balance between computational efficiency and model robustness. Additionally, we performed a set of ablation studies to assess the individual contributions of each component. Due to space constraints, detailed experimental procedures are provided in Appendix C. The source code for this project, aimed at fostering transparency and reproducibility, is available at the following URL: `https://anonymous.4open.science/r/DecVFAL-0F5C/`.

## 6.1 EXPERIMENT SETUPS

**Datasets.** Real-world VFL datasets are proprietary and not publicly accessible. Therefore, we utilized two public datasets instead for our main experiments: MNIST LeCun et al. (1998) and CIFAR-10 Krizhevsky (2009). These datasets were vertically partitioned among all participants, with each client retaining a portion of features for each sample, while the server exclusively held the labels. Detailed information about the dataset partitioning can be found in Section C.1.

**Baselines.** We deploy the baseline algorithms and DecVFAL in a hybrid cascaded VFL framework, synchronous VFL-CZOFO Wang et al. (2024). The implemented AT algorithms include PGD-$r$ Madry et al. (2017), FreeAT-$r$ Shafahi et al. (2019), FreeLB-$r$ Zhu et al. (2019), and YOPO-$m$-$n$ Zhang et al. (2019). Additionally, we integrated data parallelism, model parallelism, and asynchronous mechanisms with PGD, resulting in DP-PGD, MP-PGD, and Asy-PGD, respectively.

**Adversarial attack.** Following the threat model of adversarial attack in VFL (Appendix A.3), we employ various adversarial attack methods including FGSM Kurakin et al. (2016), PGD-$r$ Madry et al. (2017), and CW Carlini & Wagner (2017). We also simulate scenarios where malicious clients cannot directly obtain gradients and implement CERTIFY (CER) Cohen et al. (2019), zero-order-based FGSM (ZO-FGSM) and PGD (ZO-PGD) Chen et al. (2017). Additionally, Considering the case of the third-party adversary, we employ adversarial attacks that corrupt embeddings using different corrupted client selection methods: Thompson Sampling with Empirical Maximum Reward (E-TS) Duanyi et al. (2023) and All Corruption Pattern (ALL).

**Training procedures.** For the experiment applying the split MLP model on MNIST, a batch size of 32 was utilized. For the experiment applying the ResNet-18 on CIFAR-10, a batch size of 80 was used. The models were trained to converge. To ensure a fair comparison, we employed the Adam optimizer with a fixed learning rate across all VFL frameworks. Detailed parameter settings and hardware specifications for the training procedures are summarized in Appendix C.3 and Table 10.

## 6.2 EVALUATION ON ROBUSTNESS

**MNIST:** We maintain the VFL setup with one server and two clients. The server model is a single-layer perceptron, while each client employs a two-layer perceptron. The entire model is partitioned into three modules, each containing one layer. DecVFAL stands out by demonstrating the most optimal trade-off between computational efficiency and model robustness. As shown in Table 1, DecVFAL achieves the best robust performance while requiring only 1/10 of the training time per epoch for PGD adversarial training.

Table 1: Results of MNIST Robust Training

| Training Methods | Clean Accuracy | White-Box Adv. Atk | | | Black-Box Adv. Atk | | | Third Adv. | | Train Time (s/epoch) |
|---|---|---|---|---|---|---|---|---|---|---|
| | | FGSM | PGD | CW | CER | ZO-FGSM | ZO-PGD | ALL | E-TS | |
| None | 96.46 | 47.75 | 8.58 | 56.75 | 56.37 | 55.97 | 60.44 | 36.01 | 40.27 | **106.86** |
| PGD | 92.31 | 74.90 | 57.85 | 90.09 | 88.92 | 87.30 | 83.37 | 36.71 | 41.23 | 3484.57 |
| FreeAT | 92.68 | 67.29 | 41.33 | 85.11 | 84.13 | 83.51 | 80.18 | 19.01 | 20.82 | 853.64 |
| FreeLB | 93.77 | 57.18 | 18.76 | 85.30 | 82.47 | 82.30 | 79.73 | 65.33 | 71.11 | 3459.81 |
| YOPO | 96.13 | 86.36 | 73.52 | 92.49 | 91.63 | 91.17 | 88.06 | 79.81 | 84.84 | 629.43 |
| DP-PGD | 93.28 | 78.64 | 60.97 | 88.40 | 86.60 | 86.49 | 82.84 | 51.72 | 56.68 | 3451.44 |
| MP-PGD | 93.11 | 75.23 | 48.98 | 78.82 | 76.65 | 76.28 | 76.19 | 48.11 | 54.67 | 3423.91 |
| Asy-PGD | 91.25 | 72.40 | 50.41 | 84.53 | 82.55 | 82.10 | 79.50 | 38.42 | 42.99 | 3724.47 |
| DecVFAL | **98.26** | **91.62** | **77.68** | **92.84** | **91.91** | **92.13** | **89.21** | **92.20** | **94.53** | **355.16** |

**CIFAR-10:** For CIFAR10 dataset, the server model is a single-layer perceptron, whereas each client utilizes ResNet-18. For each client, the ResNet-18 model is divided into two modules: the first layer and the remaining layers. Consequently, the entire model is partitioned into three modules: the first layers of the client models, the remaining layers of the client models, and the server's single-layer perceptron. As shown in Table 2, DecVFAL achieves comparable robust performance under most of adversarial attacks while requiring only 1/3 of the training time per epoch for PGD.

Table 2: Results of CIFAR-10 Robust Training

| Training | Clean | White-Box Adv. Atk | | | Black-Box Adv. Atk | | | Third Adv. | | Train Time |
| Methods | Accuracy | FGSM | PGD | CW | CER | ZO-FGSM | ZO-PGD | ALL | E-TS | (s/epoch) |
|---|---|---|---|---|---|---|---|---|---|---|
| None | **83.93** | 53.32 | 55.42 | 62.59 | 50.39 | 52.38 | 55.58 | 76.06 | **78.93** | **70.03** |
| PGD | 78.00 | 59.08 | 68.47 | 76.73 | 70.00 | 70.32 | 70.56 | 69.54 | 72.67 | 296.35 |
| FreeAT | 80.09 | 63.63 | 61.93 | **77.01** | 68.99 | 70.99 | 71.85 | 71.44 | 74.86 | 252.11 |
| FreeLB | 81.58 | 52.09 | 54.91 | 63.70 | 53.91 | 56.92 | 59.17 | **76.30** | 78.70 | 301.43 |
| YOPO | 75.34 | 58.80 | 68.11 | 74.68 | 70.10 | 69.97 | 69.96 | 64.38 | 69.05 | 297.45 |
| DP-PGD | 75.47 | 59.37 | 68.24 | 74.56 | 69.79 | 69.74 | 70.04 | 66.19 | 69.42 | 331.93 |
| MP-PGD | 74.92 | 59.38 | 68.14 | 74.30 | 69.92 | 69.53 | 69.90 | 64.70 | 68.66 | 334.48 |
| Asy-PGD | 73.32 | 57.00 | 66.61 | 72.48 | 67.56 | 67.93 | 67.83 | 63.36 | 67.83 | 331.45 |
| DecVFAL | **81.83** | **63.69** | **68.59** | 74.72 | **71.31** | **71.05** | **72.07** | 74.93 | 77.75 | **98.99** |

### 6.3 EVALUATION ON COMPUTATIONAL EFFICIENCY

For each dataset, we trained models to converge and plotted training and testing curves in Figures 3 and 4. DecVFAL achieved better test accuracy than other baseline algorithms in significantly less time on MNIST. Due to setting close parameters to specify the number of full propagations (Table 8) for CIFAR10, DecVFAL achieved a convergence speed comparable to FreeAT and FreeLB, while delivering better robustness, as shown in Table 2.

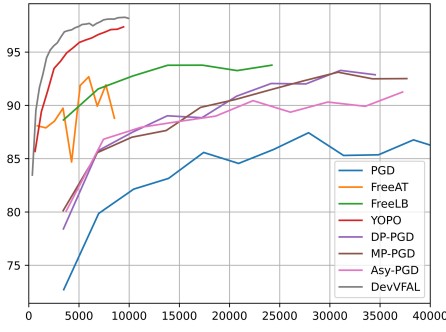
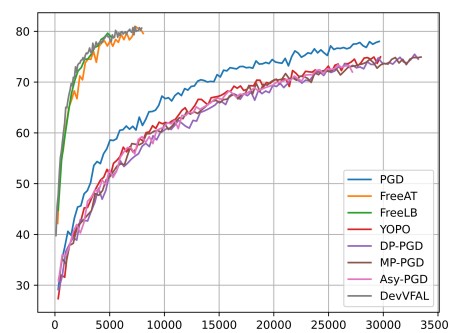

Figure 3: Training-testing curves for MNIST   Figure 4: Training-testing curves for CIFAR10

### 6.4 ABLATION STUDY

**Impact of the number of clients.** To further demonstrate the scalability of our framework, we conducted additional experiments on the MNIST dataset by varying the number of clients among 3, 5, and 7. DecVFAL consistently achieved superior robustness and enhanced computational efficiency across all client configurations compared to baseline methods. Additionally, in the scenario with 7 clients, we evaluated DecVFAL and baseline methods under third-party adversarial attacks involving corruption pattern selection, as well as attacks where some clients are malicious (as detailed in Appendix C.6). DecVFAL maintained its superior performance under these adversarial conditions.

**Limitation of the setting of $M$ and $N$.** We conducted extensive experiments on the MNIST dataset to explore the dependence on parameters $M$ and $N$. Figure 5 and Figure 6 illustrate the change in

Table 3: Results for different number of clients

| No. Clients | Training Methods | Clean Accuracy | White-Box Adv. Atk | | | Black-Box Adv. Atk | | | Third Adv. | Train Time |
|---|---|---|---|---|---|---|---|---|---|---|
| | | | PGD | FGSM | CW | CER | ZO-FGSM | ZO-PGD | ALL | (s/epoch) |
| 3 | PGD | 98.05 | 64.56 | 82.83 | 96.02 | 96.46 | 93.98 | 94.72 | 89.56 | 1015.8 |
| 5 | PGD | 96.78 | 69.50 | 84.51 | 93.00 | 95.20 | 92.36 | 93.08 | 78.62 | 1145.63 |
| 7 | PGD | 96.18 | 63.86 | 79.96 | 90.96 | 93.30 | 90.37 | 94.12 | 69.36 | 1158.11 |
| 3 | DecVFAL | 98.67 | 80.82 | 89.50 | 97.28 | 97.90 | 96.03 | 96.97 | 93.83 | 88.29 |
| 5 | DecVFAL | 98.3 | 76.52 | 87.39 | 97.34 | 97.54 | 93.85 | 95.73 | 91.87 | 92.93 |
| 7 | DecVFAL | 96.84 | 76.57 | 87.17 | 83.90 | 96.21 | 93.23 | 90.80 | 81.21 | 94.83 |

accuracy with a fixed $M = 5$ and $M = 10$, respectively, while varying $N$. It is evident that the performance rapidly degrades with increasing $N$ beyond a certain threshold, as analyzed by Corollary 1. This observation underscores the sensitivity of the model's performance to $N$, highlighting the necessity of optimizing $N$ to maintain high accuracy.

**Impact of the number of modules.** We conducted additional experiments on the MNIST dataset to evaluate how the number of partitioned modules affects the algorithm's performance. The server model was kept as a single-layer perceptron. Each client employed a ResNet-18 model, which was partitioned into varying numbers of modules: 2, 3, 4, 5, and 6. As indicated by Lemma 1, increasing the number of modules leads to larger errors in the gradient of $\eta$, which in turn negatively impacts the algorithm's accuracy. This effect is demonstrated by the results shown in Table 4.

Table 4: Results of diverse number of modules

| | Robust Accuracy (%) | | |
|---|---|---|---|
| Split Positions for Modules | Clean | FGSM | PGD |
| [: 1 : 18 : 19] | 98.90 | 48.79 | 57.49 |
| [: 1 : 9 : 18 : 19] | 98.71 | 45.88 | 55.55 |
| [: 1 : 9 : 13 : 18 : 19] | 98.58 | 44.32 | 53.42 |
| [: 1 : 5 : 9 : 13 : 18 : 19] | 98.69 | 47.09 | 45.49 |
| [: 1 : 5 : 9 : 13 : 17 : 18 : 19] | 98.22 | 38.44 | 40.63 |

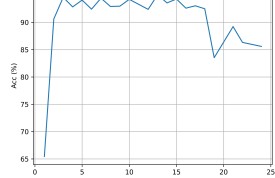 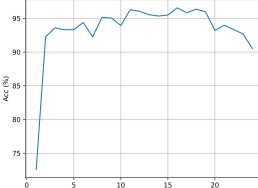

Figure 5: $M = 5$, varying $N$   Figure 6: $M = 10$, varying $N$

**Impact of split position.** We conducted additional experiments on the MNIST dataset to evaluate the effect of different split positions. The server model was kept as a single-layer perceptron, while each client utilized a ResNet-18 model that was split at various positions. The results in Table 5 demonstrate that DecVFAL performs well across various split positions compared to PGD. However, as more layers are included in the bottom module during lazy sequential backpropagation, the computational load increases, leading to longer training time.

Table 5: Results of different split positions

| Split Positions | Robust Accuracy (%) | | | Train Time |
|---|---|---|---|---|
| [: $\mathcal{M}_1$ : $\mathcal{M}_2$ : $\mathcal{M}_3$] | Clean | FGSM | PGD | (s/epoch) |
| [: 1 : 18 : 19] | 98.90 | 48.79 | 57.49 | 107.545 |
| [: 5 : 18 : 19] | 98.77 | 43.03 | 42.98 | 226.765 |
| [: 9 : 18 : 19] | 98.75 | 41.33 | 49.77 | 318.122 |
| [: 13 : 18 : 19] | 98.83 | 39.73 | 43.46 | 431.149 |
| [: 17 : 18 : 19] | 98.43 | 36.36 | 45.88 | 538.652 |
| PGD | 98.48 | 32.53 | 41.93 | 575.458 |

## 7    CONCLUSIONS

This paper presented DecVFAL, a framework that significantly accelerates VFAL while maintaining robustness. DecVFAL incorporates a dual-level decoupled mechanism to enable lazy sequential and decoupled parallel backpropagation for adversarial example generation. This approach achieves 3-10 fold speedup on MNIST and CIFAR-10 datasets, with theoretical guarantees of $\mathcal{O}(1/\sqrt{\mathcal{K}})$ convergence rate. Comprehensive experiments demonstrate DecVFAL's effectiveness across various neural architectures and VFL configurations.

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

# A BACKGROUND

## A.1 VERTICAL FEDERATED LEARNING

VFL encompasses a range of architectural designs tailored for collaborative machine learning across multiple parties. These architectures, distinguished by data and parameter distribution, as well as the trainability of the server model, include Aggregated Vertical Federated Learning ($aggVFL$) Fu et al. (2022); Liu et al. (2021b), where client parties contribute intermediate results aggregated through a non-trainable function in the server party; Aggregated Vertical Federated Learning with Central Features ($aggVFL_c$), similar to aggVFL but incorporating its own features; Split Vertical Federated Learning ($splitVFL$) Fu et al. (2022); Jin et al. (2021); Liu et al. (2021a), featuring a trainable server model processes intermediate results from passive parties; and Split Vertical Federated Learning without Local Features ($splitVFL_c$), where the server party doesn't provide any features to the model but relies solely on intermediate results from client parties.

Because VFL is a collaboration system that requires parties to exchange gradient or model level information, it has been of great research interest to study communication efficiency, and data privacy protection. Various strategies are adopted to heighten communication efficiency, typically involving reducing the cost of coordination and compressing the data transmitted between parties, such as multiple client updates Zhang et al. (2022b), asynchronous coordination Li et al. (2020), one-shot communication Wu et al. (2022), and data compression Castiglia et al. (2022); Li et al. (2020). In terms of data privacy protection, VFL relies on cutting-edge technologies like Homomorphic Encryption (HE) Yang et al. (2019), Multi-Party Computation (MPC) Xie et al. (2022); Liu et al. (2020), and Differential Privacy (DP) Wang et al. (2024) to preserve data privacy.

## A.2 VERTICAL FEDERATED ADVERSARIAL LEARNING

Emerging research has investigated the distinct challenges posed by adversarial attacks in the context of VFL Huang et al. (2024). Due to the distributed nature, VFL struggles to ensure client trustworthiness and thus renders it highly susceptible to adversarial perturbations, underscoring the pressing need for enhanced VFL model robustnessHuang et al. (2024), this is particularly evident in neural network models. Prior works have proposed that adversaries (third-party or client party) can generate adversarial samples by introducing manipulated perturbations to raw data or embeddings in the corrupted clients, aiming to mislead the inference of VFL models Luo et al. (2021); Weng et al. (2020); Qiu et al. (2022); Fu et al. (2022). However, existing VFL defense mechanisms based on cryptographic Liu et al. (2021b) and non-cryptographic Liu et al. (2021a) only concentrate on mitigating inference attacks and backdoor attacks while neglecting adversarial attacks.

## A.3 THREAT MODEL

In the context of VFL, we focus on untargeted adversarial attacks, constructed during the inference phase. The adversary's objective is to corrupt samples whose original prediction is $y_u$, causing the server model to output $\hat{y} \neq y_u$. We categorize these adversarial attacks into two primary scenarios:

- **Malicious (colluding) clients.** In this scenario, we consider the presence of malicious (colluding) clients acting as adversary. During the attack, all malicious clients (one or more) collaboratively and simultaneously generate the adversarial feature partition. The attacks are further classified based on the level of knowledge these clients possess:
    - *White-box adversarial attack.* Under relaxed protocol, clients have access to the server model $f_s$ and the output of all clients $x_{t_c}$. This protocol could occur when the client needs to make interpretable decisions based on the server model's parameters Luo et al. (2021); Wang (2019); Lundberg & Lee (2017). This implies the malicious clients have the necessary information to calculate the partial gradient to the features.
    - *Black-box adversarial attack.* Under basic VFL protocol, all participants keep their private data (e.g., labels and features), as well as the server model $f_s$ and client models $\{f_{(c)}\}_{c=1}^C$ local during inference. Clients can only receive the final prediction results $\hat{y}$ and cannot directly obtain the gradient, thus necessitating the use of black-box methods to approximate it.

- **Third party adversary.** We also consider an adversary as a third party in VFL inference, who can access, replay, and manipulate messages on the communication channel between two endpoints, where embeddings and predictions are exchanged. Third-party adversaries usually cannot achieve access to top model parameters, thus this scenario generally falls under the black-box attack category. Due to resource constraints, previous work assumed that the adversary can corrupt at most $C_a \leq C$ clients Duanyi et al. (2023).

## A.4 Adversarial Training as a Dynamical System

With the link between optimal control and deep learning Li & Hao (2018), research recast neural networks as dynamical systems and formulated the robust optimization problem as an optimal control problem Seidman et al. (2020):

$$\min_{\Theta^1,\dots,\Theta_T} \max_{\eta_1,\dots,\eta_S} \quad \sum_{i=1}^{S} \mathcal{L}(x_{T,i}, y_i) + \sum_{i=1}^{S}\sum_{t=0}^{T-1} R_t(x_{t,i}, \Theta_t)$$

$$\text{subject to} \quad x_{t+1,i} = f^t(x_{t,i}, \Theta_t), \quad i = 1,\dots,\mathcal{S}, \quad t = 1,\dots,T-1$$

$$x_{1,i} = f_0(x_{0,i} + \eta_i, \Theta_0), \quad i = 1,\dots,\mathcal{S}$$

(A.1)

where $x_t \in \mathbb{R}^{d_t}$ represents the states (i.e., the input of the $t$-th layer), $f^t : \mathbb{R}^{d_t} \times \Theta^t \to \mathbb{R}^{d_{t+1}}$ is the state transition map, $\Theta^t$ are the trainable control parameters, $\Theta$ denotes the concatenation of $(\Theta^t)_{0 \leq t \leq T-1}$, and the initial conditions are provided by the inputs to the network, $x_{0,i}$. According to the two-player Pontryagin Maximum principle, proved in Zhang et al. (2019), we define the Hamiltonians: $H_0(x, p, \theta, \eta) := p^T f_0(x + \eta, \theta) - R_0(x, \theta)$ and $H_t(x, p, \theta) := p^T f_t(x, \theta) - R_t(x, \theta)$, then there exists an optimal costate trajectory $p_t^*$, satisfied:

$$x_{t+1}^* = \nabla_p \mathcal{H}_t(x_t^*, p_{t+1}^*, \theta_t^*) \quad x_0^* = x_0 + \eta^*$$

(A.2)

$$p_t^* = \nabla_x \mathcal{H}_t(x_t^*, p_{t+1}^*, \theta^{t,*}) \quad p_T^* = -\nabla \mathcal{L}(x_T^*, y)$$

(A.3)

where $\Theta^* := \{\theta^{0,*}, \dots \theta^{T-1,*}\}$ is the solution of the problem (A.1).

Due to the compositional structure, feed-forward deep neural networks can be viewed as dynamical systems. This approach has been recently explored in several papers, which leverage this interpretation to propose new training algorithms (Weinan, 2017; Li et al., 2018; Weinan et al., 2018; Zhang et al., 2019).

According to equation A.1, the two-player Pontryagin Maximum principle, proved in (Zhang et al., 2019), gives necessary conditions for an optimal setting of the parameters $\theta^*$, perturbations $\eta_1^*, \dots, \eta_S^*$, and corresponding trajectories $\{x_{t,i}^*\}$. Define the Hamiltonians

$$H_t(x, p, \theta) := p^\top f_t(x, \theta) - R_t(x, \theta), \quad t = 1, \dots, T-1$$

$$H_0(x, p, \theta, \eta) := p^\top f_0(x + \eta, \theta) - R_0(x, \theta)$$

(A.4)

The two-player maximum principle says in this case that if $\Phi$, $f_t$, and $R_t$ are twice continuously differentiable, with respect to $x$, uniformly bounded in $x$ and $t$ along with their partial derivatives, and the image sets $\{f_t(x, \theta) | \theta \in \mathbb{R}^{m_t}\}$ and $\{R_t(x, \theta) | \theta \in \mathbb{R}^{m_t}\}$ are convex for all $x$ and $t$, then there exists an optimal costate trajectory $p_t^*$ such that the following dynamics are satisfied

$$x_{t+1,i}^* = \nabla_p H_t(x_{t,i}^*, p_{t+1,i}^*, \theta_t^*), \quad x_{1,i}^* = \nabla_p H_0(x_{0,i}, p_{1,i}^*, \theta_0^*, \eta_i^*)$$

$$p_{t,i}^* = \nabla_x H_t(x_{t,i}^*, p_{t+1,i}^*, \theta_t^*), \quad p_{T,i}^* = -\nabla_x \Phi(x_{T,i}^*, y_i)$$

(A.5)

and the following Hamiltonian condition for all $\theta_t \in \mathbb{R}^{m_t}$ and $\eta_i \in X$

$$H_t(x_{t,i}^*, p_{t+1,i}^*, \theta_t) \leq \sum_{i=1}^{S} H_t(x_{t,i}^*, p_{t+1,i}^*, \theta_t^*), \quad t = 1, \dots, T-1$$

$$\sum_{i=1}^{S} H_0(x_{t,i}^*, p_{t+1,i}^*, \theta_t, \eta_i^*) \leq \sum_{i=1}^{S} H_0(x_{t,i}^*, p_{t+1,i}^*, \theta_t^*, \eta_i^*) \leq \sum_{i=1}^{S} H_0(x_{t,i}^*, p_{t+1,i}^*, \theta_t^*, \eta_i)$$

(A.6)

These necessary optimality conditions can be used to design an iterative algorithm of the following form. For each data point $i \in \{1, \dots, S\}$,

1. Compute the state and costate trajectories $\{x_{i,t}\}$ and $\{p_{i,t}\}$ from the dynamics, keeping $\theta_t$ and $\eta_i$ fixed:

$$x_{t+1,i}^{(\eta)} = \nabla_p H_t(x_{t,i}^{(\eta)}, p_{t+1,i}^{(\eta)}, \theta_t)$$
$$x_{1,i}^{(\eta)} = \nabla_p H_0(x_{0,i}, p_{1,i}^{(\eta)}, \theta_0, \eta)$$

2. $p_{t,i}^{(\eta)} = \nabla_x H_t(x_{t,i}^{(\eta)}, p_{t+1,i}^{(\eta)}, \theta_t), p_{T,i}^{(\eta)} = -\nabla_x \Phi(x_{T,i}^{(\eta)}, y_i)$

3. Minimize the Hamiltonian $H_0(x_t, i, pt+1, i, \theta_t, \eta_i)$ with respect to $\eta_i$

4. Maximize the sum of Hamiltonians $\sum_{i=1}^{S} H_t(x_t, i, pt+1, i, \theta_t)$ with respect to $\theta_t$ for all $t$

As was noticed as early as (LeCun et al., 1988), it can be seen from the chain rule that the backward costate dynamics are equivalent to backpropagation through the network. With this interpretation, the gradient of the total loss for the $i$-th data point with respect to the adversary $\eta_i$ can be written as $\nabla_\eta f_0(x_{0,i} + \eta_i, \theta_0)^\top p_{1,i}^{(\eta)}$. For a fixed value of $\theta_0$, performing gradient descent on $H_0$ to find a worst-case adversarial perturbation can be expressed as the following updates, where $\alpha > 0$ is a step size:

$$\eta_i^{(\ell+1)} = \eta_i^{(\ell)} - \alpha \nabla_\eta f_0(x_{0,i} + \eta_i^{(\ell)}, \theta_0)^\top p_{1,i}^{(\eta)} \tag{A.7}$$

An important observation made in (Zhang et al., 2019) is that the adversary is only present in the first layer Hamiltonian condition and this function can be minimized by computing gradients only with respect to the first layer of the network. More explicitly, instead of using $p_{\ell,1}^{(\eta)}$, as in the updates above, we could instead use $p_{0,1}^{(\eta)}$ and the updates

$$\eta_i^{(\ell+1)} = \eta_i^{(\ell)} - \alpha \nabla_\eta f_0(x_{0,i} + \eta_i^{(\ell)}, \theta_0)^\top p_{0,1}^{(\eta)} \tag{A.8}$$

This removes the need to do a full backpropagation to recompute the costate $p_{\ell,1}^{(\eta)}$ for every update of $\eta_i^{(\ell)}$, at the cost of now being an approximate gradient.

### A.5 ZEROTH ORDER OPTIMIZATION

ZOO methods Huang et al. (2020; 2019) have been developed to effectively solve many ML problems for which obtaining explicit gradient expressions is difficult or infeasible. Such problems include structure prediction tasks, where explicit gradients are challenging to derive Sokolov et al. (2018), as well as bandit and black-box learning problems Shamir (2017); Liu et al. (2018), where obtaining explicit gradients is not feasible. Specifically, ZOO relies solely on function values for optimization, eschewing the need for explicit gradients.

Formally, given a function $f(x)$ with input $x$, the gradient $\nabla f(x)$ can be estimated using ZOO. One common approach is to sample random perturbations $u$ within the domain of $f$ and evaluate the function shifts. The ZO gradient estimator $\hat{\nabla} f(x)$ is given by:

$$\hat{\nabla} f(x) = \frac{1}{q} \sum_{j=1}^{q} [f(x + \mu u_j) - f(x)] \frac{u_j}{\mu} \tag{A.9}$$

where $\mu$ serves as a scaling factor for the random perturbation, while $u_j$ represents the $j$-th random perturbation sampled from a distribution $p$ across the domain of $f$. The parameter $q$ denotes the number of random samples employed for estimation. Normalizing the perturbation by $\frac{u_j}{\mu}$ ensures the estimator's unbiasedness. The expectation of the Zeroth Order (ZO) gradient estimator yields an unbiased estimate of the true gradient, expressed as $E[\hat{\nabla} f(x)] = \nabla f(x)$, provided that the samples $u_j$ are drawn from a distribution with a mean of zero.

The application of ZOO to VFL has been discussed, highlighting its specific properties such as model applicability Zhang et al. (2021), privacy security concerns Liu et al. (2018), and considerations regarding communication cost and computational efficiency Wang et al. (2024).

## A.6 COMMUNICATION COMPRESSION

Compression is a pivotal technique in VFL that aims to mitigate communication overhead by reducing the volume of data transmitted among participating parties. In the context of neural network-based VFL algorithms, high-dimensional input vectors are inherently mapped onto lower-dimensional representations, which serve a natural compression purpose. However, to further enhance communication efficiency, specialized dimensionality reduction techniques are often integrated. Several VFL frameworks have been proposed to incorporate compression techniques: AVFL Cai et al. (2024) leverages PCA to compress the data before transmission, reducing the communication load. CE-VFL Khan et al. (2022) employs both PCA and autoencoders to learn latent representations from raw data, which are then used for model training. SecureBoost+ Chen et al. (2021b) and eHE-SecureBoost Xu et al. (2021) encode encrypted gradients into a compact form, minimizing the number of cryptographic operations and the data transmission size. C-VFL Castiglia et al. (2022) introduces an arbitrary compression scheme to VFL, offering a theoretical analysis of how compression parameters impact the overall system efficiency.

Compression techniques play a critical role in VFL by enabling more efficient data transmission without compromising the integrity of the learning process. The selection of an appropriate compression method is contingent upon the specific requirements of the VFL scenario, including the sensitivity of the data, the computational resources available, and the desired balance between communication efficiency and model performance.

# B CONVERGENCE ANALYSIS

## B.1 NOTATIONS

| Notations | Definitions |
|---|---|
| **Neural Network Classifier** | |
| $\mathcal{S}$ | The number of samples |
| $f$ | Neural network model |
| $\Theta$ | Model Parameters |
| $x_i, y_i$ | Input sample and corresponding label |
| $\mathcal{B}, B$ | The mini-batch $\mathcal{B}$ with size $B$ |
| $\mathbb{E}$ | Expectation |
| $k \in \{1, 2, \ldots, \mathcal{K}\}$ | Iteration index for parameter updating |
| **Vertical Federated Learning** | |
| $C$ | The number of clients |
| $f_{(1)}, f_{(2)}, \ldots, f_{(C)}$ | Client models |
| $\theta = \{\theta_{(1)}, \theta_{(2)}, \ldots, \theta_{(C)}\}$ | Client model parameters |
| $f_s$ | Server model |
| $\psi$ | Server model parameters |
| $\mathcal{L}$ | Loss function |
| $f = \{f_s, f_{(1)}, f_{(2)}, \ldots, f_{(C)}\}$ | The complete federated model |
| $\alpha_\psi$ | Learning rate for server model parameters |
| $\alpha_\theta$ | Learning rate for client model parameters |
| **Adversarial Training** | |
| $\mathcal{A}$ | Adversarial Loss Function |
| $\mathcal{G}_\mathcal{B}(\eta, \psi, \theta)$ | $\frac{1}{B} \sum_{i \in \mathcal{B}} \mathcal{A}_i(\eta_i, \psi_i, \theta_i)$ |
| $\mathcal{R}(\eta, \psi, \theta)$ | $\frac{1}{\mathcal{S}} \sum_{i \in \mathcal{S}} \mathcal{A}_i(\eta_i, \psi_i, \theta_i)$ |
| $\eta_i^*$ | $\mathrm{argmax}_\eta \mathcal{A}_i(\eta, \psi, \theta)$ |
| $\eta$ | Adversarial perturbation |
| $\Pi$ | Projection operator |
| $\alpha_\eta$ | Learning rate for adversarial sample |
| $\ell$ | Iteration index for adversarial sample generation |
| $x_{0,i} = \{x_{0,i,(1)}, x_{0,i,(2)}, \ldots, x_{0,i,(C)}\}$ | The sample $i$ from all clients |
| $\eta_i = \{\eta_{i,(1)}, \eta_{i,(2)}, \ldots, \eta_{i,(C)}\}$ | the adversarial perturbation for sample $i$ |
| **Optimal Control Formulation of Deep Learning** | |
| $\mathcal{H}_t$ | Hamiltonian function for layer $t$ |
| $p_t = \{p_{t,(1)}, p_{t,(2)}, \ldots, p_{t,(C)}\}$ | Costates at layer $t$ |
| $T$ | Number of layers in the neural network |
| $t = 0, 1, \ldots, T-1$ | Layer indices |
| $f^t$ | State transition map for layer $t$ |
| $x_t = \{x_{t,(1)}, x_{t,(2)}, \ldots, x_{t,(C)}\}$ | States at layer $t$ |
| $\Theta^t$ | Trainable parameters for layer $t$ |

Table 6: Table of Notations

| Notations | Definitions |
|---|---|
| **Decoupled parallel Backpropagation** | |
| $\mathcal{M}_K$ | The number of divided modules |
| $t_s$ | The number of server model's layers |
| $t_c$ | The number of client model's layers |
| $f = \{f_1, f_2, \ldots, f_{t_c}, \ldots, f_{T-1}\}$ | Classifier from layer-wise view |
| $\theta = \{\Theta_1, \Theta_2, \ldots, \Theta_{t_c}\}$ | Client model parameters from layer-wise view |
| $x_{t_c}$ | The output of all clients |
| $f_{\tilde{\theta}_1}$ | Client model network excluding the first layer |
| **Lazy Sequential Backpropagation** | |
| $M$ | Number of iterations for full propagations |
| $N$ | Number of iterations for propagations in bottom module |
| $R_t$ | Regularizer for layer $t$ |
| $f_{\tilde{\Theta}_1}$ | Network excluding the first layer |
| $x_{t,i}^{m,n}$ | The state of sample $i$ at layer $t$ in $m, n$ iteration |
| $p_{t,i}^{m,n}$ | The co-state of sample $i$ at layer $t$ in $m, n$ iteration |
| **Zeroth Order Gradient Estimation** | |
| $\mu$ | Smoothing parameter |
| $\mathbf{u}$ | Random vector |
| $q$ | Query budget for gradient estimation |
| $\{\delta_i^j\}_{j=1}^q$ | Loss difference |
| $\hat{\nabla}\mathcal{A}(\eta, \psi, \theta)$ | Estimation Gradient from ZOO |
| **Compressor** | |
| $\mathcal{C}(\cdot)_b$ | Compressor compressing information to $b$ bits |
| $\nabla\hat{\mathcal{A}}(\eta, \psi, \theta)$ | Compression Gradient |

Table 7: Table of Notations (continue)

### B.2 Assumptions

**Assumption 1.** *Lipschitz Gradient: There exists a constant $K > 0$ such that for all $t \in 1, \ldots, t_c, \ldots, T$, the functions $f_t$, $f_c$, $\mathcal{L}$, and $R_t$ are $K$-Lipschitz in $x$, uniformly in $\theta$ and $\psi$. For all each sample $i \in 1, \ldots, \mathcal{S}$, the function $\nabla_\eta \mathcal{A}_i(\eta, \psi, \theta)$, $\nabla_\psi \mathcal{A}_i(\eta, \psi, \theta)$, $\nabla_\theta \mathcal{A}_i(\eta, \psi, \theta)$ satisfy the following Lipschitz conditions:*

$$||\nabla_\eta \mathcal{A}_i(\eta, \psi', \theta) - \nabla_\eta \mathcal{A}_i(\eta, \psi, \theta)|| \leq L_{\eta\psi}||\psi' - \psi|| \tag{B.1}$$

$$||\nabla_\eta \mathcal{A}_i(\eta, \psi, \theta') - \nabla_\eta \mathcal{A}_i(\eta, \psi, \theta)|| \leq L_{\eta\theta}||\theta' - \theta|| \tag{B.2}$$

$$||\nabla_\psi \mathcal{A}_i(\eta', \psi, \theta) - \nabla_\psi \mathcal{A}_i(\eta, \psi, \theta)|| \leq L_{\psi\eta}||\eta' - \eta|| \tag{B.3}$$

$$||\nabla_\psi \mathcal{A}_i(\eta, \psi, \theta') - \nabla_\psi \mathcal{A}_i(\eta, \psi, \theta)|| \leq L_{\psi\theta}||\theta' - \theta|| \tag{B.4}$$

$$||\nabla_\theta \mathcal{A}_i(\eta', \psi, \theta) - \nabla_\theta \mathcal{A}_i(\eta, \psi, \theta)|| \leq L_{\theta\eta}||\eta' - \eta|| \tag{B.5}$$

$$||\nabla_\theta \mathcal{A}_i(\eta, \psi', \theta) - \nabla_\theta \mathcal{A}_i(\eta, \psi, \theta)|| \leq L_{\theta\psi}||\psi' - \psi|| \tag{B.6}$$

**Assumption 2.** *Unbiased Gradient and Bounded Variance: There exists $\sigma_\psi > 0$ and $\sigma_\theta > 0$, the stochastic gradients are unbiased, i.e. $\mathbb{E}_i \nabla_\psi \mathcal{G}_i(\eta, \psi, \theta) = \nabla_\psi \mathcal{R}(\eta, \psi, \theta), \mathbb{E}_i \nabla_\theta \mathcal{G}_i(\eta, \psi, \theta) = \nabla_\theta \mathcal{R}(\eta, \psi, \theta), i = 1, \ldots, B$ and satisfy:*

$$\mathbb{E}||\nabla_\psi \mathcal{G}_\mathcal{B}(\eta, \psi, \theta) - \nabla_\psi \mathcal{R}(\eta, \psi, \theta)||^2 \leq \sigma_\psi^2 \tag{B.7}$$

$$\mathbb{E}||\nabla_\theta \mathcal{G}_\mathcal{B}(\eta, \psi, \theta) - \nabla_\theta \mathcal{R}(\eta, \psi, \theta)||^2 \leq \sigma_\theta^2 \tag{B.8}$$

Assumption 1, 2 are the basic assumptions for solving the non-convex optimization problem with stochastic gradient descentWang et al. (2023)Haddadpour & Mahdavi (2019).

**Assumption 3.** *Bounded Hessian: The Hessian for $\mathcal{A}_i(\eta, \psi, \theta)$ is bounded, i.e.there exist positive constants $H_\psi$ and $H_\theta$ for $\mathcal{A}_i(\eta, \psi, \theta)$, $\psi$ and $\theta$, the following inequalities holds:*

$$||\nabla_\psi^2 \mathcal{A}_i(\eta_i, \psi, \theta)|| \leq H_\psi \tag{B.9}$$

$$||\nabla_{[\theta, x_{0,i}]}^2 \mathcal{A}_i(\eta_i, \psi, \theta)|| \leq H_\theta \tag{B.10}$$

**Assumption 4.** *Bounded Block-coordinate Gradient: The gradient of all the participants' local output w.r.t. their local input is bounded, i.e. for, all $i \in 1, \ldots, \mathcal{S}$ there exist positive constants $Q_\psi$ and $Q_\theta$ satisfies the following inequalities:*

$$||\nabla_{[\psi]} \mathcal{A}_i(\eta_i, \psi, \theta)|| \leq Q_\psi \tag{B.11}$$

$$||\nabla_\theta \mathcal{A}_i(\eta_i, \psi, \theta)|| \leq Q_\theta \tag{B.12}$$

Assumption 3, 4 are the fundamental assumptions for bounding the compression loss. Compression introduces errors into the input of the loss function; therefore, with a bounded Hessian, we can determine the maximum effect of these errors on the loss. Additionally, bounding the block-coordinated gradient is a common practice in VFL analysis. This approach helps constrain the entire model's gradient when the gradients of other parts have been bounded Wang et al. (2024)Castiglia et al. (2022).

**Assumption 5.** *$z$-Strongly Concave: If function $\mathcal{A}_i(\eta, \psi, \theta)$ is $z$-strongly concave for $\eta$, then for all $\psi$ and $\theta$, the following inequalities satisfy:*

$$||\eta' - \eta|| \leq (1/z)||\nabla_\eta \mathcal{A}_i(\eta, \psi, \theta)|| \tag{B.13}$$

Assumption 5 made in previous results on convergence of robust training Wang et al. (2021) and is justified through the reformulation of robust training as distributionally robust optimization. It helps us to bound the delayed gradient of $\eta$.

### B.3 Proposition

**Proposition 1.** *Under **Assumption 1** and **Assumption 5**, the loss function $\mathcal{R}(\eta', \psi, \theta)$ is $L_\psi$-smooth for $\psi$, $L_\theta$-smooth for $\theta$, and the following inequality holds for all $\psi$, $\psi'$, $\theta$, and $\theta'$:*

$$\mathcal{R}(\eta', \psi', \theta') - \mathcal{R}(\eta, \psi, \theta) \leq \langle \nabla_\theta \mathcal{R}(\eta, \psi, \theta), \theta' - \theta \rangle + \frac{L_\theta}{2} ||\theta' - \theta||^2$$

$$+ \langle \nabla_\psi \mathcal{R}(\eta, \psi, \theta), \psi' - \psi \rangle + \frac{L_\psi}{2} ||\psi' - \psi||^2 \tag{B.14}$$

where $L_\psi = L_{\psi\psi} + \frac{L_{\psi\eta}L_{\eta\psi}}{z}$ and $L_\theta = L_{\theta\theta} + \frac{L_{\theta\eta}L_{\eta\theta}}{z}$. *This assumption is consistent with **Proposition 1** in Seidman et al. (2020). This can help us to connect the N-loop and M-loop.*

**Proposition 2.** *The classical back-propagation-based gradient descent algorithm can be viewed as an algorithm attempting to solve the PMPZhang et al. (2019). The costate processes $p_t^*$ and the gradient $\nabla_{x_t}\mathcal{A}(\eta, \psi, \theta)$ satisfy the following equation:*

$$p_t = -\nabla_{x_t}\mathcal{A}(\eta, \psi, \theta) \tag{B.15}$$

### B.4 DEFINITION

**Definition 1.** ***Compression Error (forward message)*** *Considering sample $i$, we can define the compression error of $\mathcal{C}(\cdot)_b$: $e_{c,i}$, $c \in 1, 2, ..., C$, i.e. $e_{c,i} = \mathcal{C}(x_{t_c,c,i})_b - x_{t_c,c,i}$. We denote the expected norm of the error from the client $c$ at global iteration $k$ as $\mathcal{E}_{c,i}^k = \mathbb{E}||e_{c,i}^k||^2$, and $\mathcal{E}^k = \max_c \mathcal{E}_{c,i}^k$. Since all client operations are synchronized, the error from all clients is $e_i^k = (e_{1,i}^k, e_{2,i}^k, ..., e_{C,i}^k)$. Then, the expected norm of the error from all clients:*

$$\mathbb{E}||e_i^k||^2 = \mathbb{E}||(e_{1,i}^k, e_{2,i}^k, ..., e_{C,i}^k)||^2$$
$$\leq \sum_{c=1}^C \mathbb{E}||e_{c,i}^k||^2$$
$$\leq C\mathcal{E}^k \tag{B.16}$$

### B.5 LEMMA

**Lemma 3.** ***Zeroth-Order Optimization.*** *For arbitrary $f$ in problem $(P)$, the following conditions hold:*
*1) $f_\mu(x)$ is continuously differentiable, its gradient is $L_\mu$-Lipschitz continuous with $L_\mu \leq L$:*

$$\nabla f_\mu(x) = \mathbb{E}_{\mathbf{u}}[\hat{\nabla}f(x)] \tag{B.17}$$

*where $\mathbf{u}$ is drawn from the uniform distribution over the unit Euclidean sphere, $f_\mu(x) = \mathbb{E}(f(x+\mu\mathbf{u}))$ is the smooth approximation of $f$.*
*2) For any $x \in \mathbb{R}^d$, the following inequalities satisfy:*

$$||\nabla f_\mu(x) - \nabla f(x)||^2 \leq \frac{L^2\mu^2d^2}{4} \tag{B.18}$$

Proof of this lemma is provided in Liu et al. (2018); Gao et al. (2018).

**Lemma 4.** ***Bound the costate difference at the bottom module.*** *There exists a constant $G'$ dependent on $T$ and $K$ such that for all $n \in \{0, \ldots, N\}$, $m \in \{0, \ldots, M\}$, and $i \in \{1, \ldots, S\}$:*

$$\left\|p_{\mathcal{M}_1,i}^{m-\tau_1,0} - p_{\mathcal{M}_1,i}^{m,N}\right\| \leq G'\alpha_\eta(\mathcal{M}_K N - 1). \tag{B.19}$$

*Where $G' = TK^{T+1}(K^T + T(T-1)K^{2T-2} + TK^{2T})$, $m$ is the iteration index of full propagation.*

**Proof:** This lemma bounds the difference of the costates of the first module in the adversary's $N$-loop. We fix the data point $i$, and for ease of notation drop the dependence of state variables on the index $i$, while also suppressing the notational dependence on $\Theta$ for all functions, as $\Theta$ is fixed during the updates for the adversary $\eta$. We define $x_t$ and $p_t$ as the state and costate trajectories generated from the initial condition $x_0 + \eta$. We additionally define $\delta p_t^\ell := p_t^0 - p_t^\ell$ and $\delta x_t^\ell := x_t^0 - x_t^\ell$, $\ell$ is the iteration index of example updates. We first prove bounds on $\|p_t^\ell\|$ and $\|\delta x_t^\ell\|$.

Applying Assumption (1), we have:

$$\|p_T^\ell\| \leq \|-\nabla\Phi(x_T^\ell, y)\| \leq K \tag{B.20}$$
$$\|p_t^\ell\| = \|\nabla_x\mathcal{H}_t(x_t^\ell, p_{t+1}^\ell, \theta_t)\|$$
$$\leq \|p_{t+1}^\ell\|\|\nabla_x f_t(x_t^\ell, \theta_t)\| + \|\nabla_x R_t(x_t^\ell)\|$$
$$\leq K\|p_{t+1}^\ell\| + K$$
$$\leq K + K^2 + \ldots + K^{T-t+1}$$
$$\leq K^{T-t+1}(T - t + 1) \tag{B.21}$$

Next, from Assumption (1), we have $\|\delta x_1^\ell\| = \|f_1(x_0 + \eta^0) - f_1(x_0 + \eta^\ell)\| \leq K\|\eta^0 - \eta^\ell\|$. By induction, we have:

$$\|\delta x_t^\ell\| \leq K^t\|\eta^0 - \eta^\ell\| \tag{B.22}$$

To bound $\|p_{\mathcal{M}_1}^0 - p_{\mathcal{M}_1}^\ell\|$, we first note that $\|\delta p_T^\ell\| = \|\nabla\Phi(x_T^\ell) - \nabla\Phi(x_T^0)\| \leq K\|\delta x_T^\ell\|$. We write:

$$
\begin{aligned}
\|\delta p_t^\ell\| &= \|\nabla_x H_t(x_t^0, p_{t+1}^0) - \nabla_x H_t(x_t^\ell, p_{t+1}^\ell)\| \\
&= \|\nabla_x H_t(x_t^0, p_{t+1}^0) - \nabla_x H_t(x_t^\ell, p_{t+1}^0) + \nabla_x H_t(x_t^\ell, p_{t+1}^0) - \nabla_x H_t(x_t^\ell, p_{t+1}^\ell)\| \\
&= \|\langle p_{t+1}^0, \nabla_x f_t(x_t^0) - \nabla_x f_t(x_t^\ell)\rangle + \langle p_{t+1}^0 - p_{t+1}^\ell, \nabla_x f_t(x_t^\ell)\rangle + \nabla_x R_t(x_t^\ell) - \nabla_x R_t(x_t^0)\| \\
&\leq K^{T-1}\left(K\|\delta x_T^\ell\| + \sum_{t=1}^{T-1}(K^{T-t+1}(T-t) + K)\|\delta x_t^\ell\|\right)
\end{aligned}
\tag{B.23}
$$

Applying (B.22), we have:

$$\|\delta p_{\mathcal{M}_1}^\ell\| \leq (K^T + T(T-1)K^{2T-2} + TK^{2T})\|\eta^0 - \eta^\ell\| \tag{B.24}$$

$\eta$ updates with the form:

$$\eta^{\ell+1} = \eta^\ell - \alpha_\eta \nabla_\eta f_{\mathcal{M}_1}(x_0 + \eta^\ell, \theta_{\mathcal{M}_1})^\top p_{\mathcal{M}_1}^0 \tag{B.25}$$

Applying Assumption (1) and (B.21), we have:

$$\|\eta^0 - \eta^\ell\| \leq K^{T+1}T\alpha_\eta(\ell - 1) \tag{B.26}$$

Finally, substituting with (B.26) gives us the desired result:

$$\|p_{\mathcal{M}_1,i}^0 - p_{\mathcal{M}_1,i}^\ell\| \leq G'\alpha_\eta(\ell - 1) \tag{B.27}$$

where $G' = TK^{T+1}(K^T + T(T-1)K^{2T-2} + TK^{2T})$.

Then, We are going to bound $\left\|p_{\mathcal{M}_1,i}^{m-\tau_1,0} - p_{\mathcal{M}_1,i}^{m,N}\right\|$:

$$
\begin{aligned}
\left\|p_{\mathcal{M}_1,i}^{m-\tau_1,0} - p_{\mathcal{M}_1,i}^{m,N}\right\| &= \left\|p_{\mathcal{M}_1,i}^{m-\tau_1,0} - p_{\mathcal{M}_1,i}^{m,0} + p_{\mathcal{M}_1,i}^{m,0} - p_{\mathcal{M}_1,i}^{m,N}\right\| \\
&\overset{(a)}{\leq} \left\|p_{\mathcal{M}_1,i}^{m-\tau_1,0} - p_{\mathcal{M}_1,i}^{m,0}\right\| + \left\|p_{\mathcal{M}_1,i}^{m,0} - p_{\mathcal{M}_1,i}^{m,N}\right\| \\
&\overset{(b)}{\leq} G'\alpha_\eta(\tau_1 N) + G'\alpha_\eta(N - 1) \\
&\leq G'\alpha_\eta[(\tau_1 + 1)N - 1] \\
&\leq G'\alpha_\eta[\mathcal{M}_K N - 1]
\end{aligned}
\tag{B.28}
$$

Here, $(a)$ is obtained using the triangle inequality, $(b)$ is obtained using (B.27), for each $M$-loop, the adversary is updated $N$ times. Proof completes.

**Lemma 5.** *Bound Compression Error. Under Assumption 3, 4, and Definition 1, the norm of the difference between the loss function value with and without compression error is bounded:*

$$\mathbb{E}\|\nabla_\psi \hat{\mathcal{A}}_i(\eta, \psi, \theta) - \nabla_\psi \mathcal{A}_i(\eta, \psi, \theta)\| \leq H_\psi^2 C\mathcal{E}^k \tag{B.29}$$

$$\mathbb{E}\|\nabla_\theta \hat{\mathcal{A}}_i(\eta, \psi, \theta) - \nabla_\theta \mathcal{A}_i(\eta, \psi, \theta)\| \leq Q_\theta^2 H_\theta^2 C\mathcal{E}^k \tag{B.30}$$

$$\mathbb{E}\|\nabla_{x_{t_c}} \hat{\mathcal{A}}_i(\eta, \psi, \theta) - \nabla_{x_{t_c}} \mathcal{A}_i(\eta, \psi, \theta)\| \leq H_\theta^2 C\mathcal{E}^k \tag{B.31}$$

The proof of this lemma proceeds same to **Lemma D.4** in Wang et al. (2024).

**Lemma 6.** *Bound the gradient for $\eta$. Due to the communication between the clients and the server involved in the update process of adversarial examples, the gradient $\nabla_\eta \mathcal{A}$ respect to $\eta$ involves estimation gradient $\nabla_\eta \hat{\mathcal{A}}$ from ZOO and compression gradient $\hat{\nabla}_\eta \mathcal{A}$. Under the Assumption 1, and*

***Lemma 3, 5***, *at the $i$-th sample and $k$-th iteration, the pseudo-partial derivative for $\eta$ satisfies the following inequality:*

$$\hat{\eta}_i = \underset{\substack{m=1,...,M \\ n=1,...,N}}{argmin} \left\| \hat{\nabla}_\eta \hat{\mathcal{A}}_i \left( \eta_i^{m,n}, \psi_i, \theta_i \right) \right\|,$$

$$\mathbb{E} \left\| \hat{\nabla}_\eta \hat{\mathcal{A}}_i (\hat{\eta}_i, \psi_i, \theta_i) \right\|^2 \le \left[ D(\mathcal{X}) L_{\eta\eta}^2 \left( 1 - \frac{z}{L_{\eta\eta}} \right)^{MN+1} + \frac{2G^2}{L_{\eta\eta}} \left[ (\mathcal{M}_K N - 1 \right]^2 \left( \frac{2}{z} + \frac{1}{2L_{\eta\eta}} \right) \right]$$

$$\times 3 \left( H_\theta^2 C \mathcal{E}^k + \frac{L^2 \mu^2 d^2}{4} + K^2 \right) \tag{B.32}$$

*where $G = KG'$, $\alpha_x < \frac{1}{L_{\eta\eta}}$, and $\eta \in \mathcal{X}$.*

**Proof:**
According to the chain rule, we note that $\hat{\nabla}_\eta \hat{\mathcal{A}}_i(\hat{\eta}_i, \psi_i, \theta_i)$ can be split as follows:

$$\mathbb{E} \left\| \hat{\nabla}_\eta \hat{\mathcal{A}}_i(\hat{\eta}_i, \psi_i, \theta_i) \right\|^2 = \mathbb{E} \| \nabla_\eta x_{t_c, i} \hat{\nabla}_{x_{t_c}} \hat{\mathcal{A}}_i(\hat{\eta}, \psi_i, \theta_i) \|^2$$

$$\le \underbrace{\mathbb{E} \| \nabla_\eta x_{t_c, i} \|^2}_{a} \underbrace{\mathbb{E} \| \hat{\nabla}_{x_{t_c}} \hat{\mathcal{A}}_i(\hat{\eta}, \psi_i, \theta_i) \|^2}_{b} \tag{B.33}$$

For (a): we view the clients' networks as an independent model. From **Proposition 2**, we can get the following:

$$\| p_{t_c, i}^{m,n} \| = \| -\nabla_{x_{t_c}} \mathcal{A}_i(\eta_i^{m,n}, \psi_i, \theta_i) \| \le K \tag{B.34}$$

Where $m = 1, 2, ..., M$ denotes $M$-loop index, $n = 1, 2, ..., N$ denotes $N$-loop index.

According to the **Lemma 8** in Seidman et al. (2020), we drop the dependence of all functions on $\Theta$ and the data point index $i$ for the proof. The $N$-loop of the adversary's updates can be written as (B.25). Recall that the true gradient of $\mathcal{A}(\eta^{m,N})$ is

$$\nabla_\eta \mathcal{A}(\eta^{m,N}) = \nabla_\eta f_{\mathcal{M}_1}(x + \eta)^\top p_{\mathcal{M}_1}^{m,N}. \tag{B.35}$$

We will bound the maximum difference of the update vector to the true gradient over the iterations of the adversary's updates. In this sense, the adversary's updates can be viewed as a standard gradient method with an inexact gradient oracle. We write

$$\| \nabla_\eta f_{\mathcal{M}_1}(x + \eta)^\top p_{\mathcal{M}_1}^{m-\tau,0} - \nabla_\eta \mathcal{A}(\eta^{m,N}) \| = \| \nabla_\eta f_{\mathcal{M}_1}(x + \eta)^\top p_{\mathcal{M}_1}^{m-\tau,0} - \nabla_\eta f_{\mathcal{M}_1}(x + \eta)^\top p_{\mathcal{M}_1}^{m,N} \|$$

$$\le \| p_{1,}^{m-\tau,0} - p_1^{m,N} \| \| \nabla_\eta f_{\mathcal{M}_1}(x + \eta)^\top \|$$

$$\le KG'\alpha_\eta \left[ (\mathcal{M}_K N - 1 \right] \tag{B.36}$$

$$= G\alpha_\eta \left[ \mathcal{M}_K N - 1 \right] \tag{B.37}$$

We now appeal to an inexact oracle convergence result in Devlin et al. (2019). Given a concave function $f(x')$ and a point $x'$, we define a $(\delta, \mu, L)$ oracle as returning a vector $g(x')$ such that the following inequality holds:

$$\frac{\mu}{2} \| x' - x \|^2 \le f(x') - f(x) + \langle g(x'), x' - x \rangle \le \frac{L}{2} \| x' - x \|^2 + \delta \tag{B.38}$$

It can be shown that if we have an approximate gradient bound of the form (B.36), and $\mathcal{A}$ is $L_{\eta\eta}$-smooth (Assumption 1) and $z$-strongly concave in $\eta$ (Assumption 5), then the updates for the adversary are created by a $(\delta, z/2, 2L_{\eta\eta})$-oracle, where

$$\delta = G^2 \alpha_\eta^2 \left[ \mathcal{M}_K N - 1 \right]^2 \left( \frac{2}{z} + \frac{1}{2L_{\eta\eta}} \right) \tag{B.39}$$

Letting $\alpha_\eta < 1/L_{\eta\eta}$ and applying Theorem 4 in Devlin et al. (2019), along with the inequality $\| \nabla A(\hat{\eta}) \|^2 \le 2L_{\eta\eta}(\max_\eta A(\eta) - A(\hat{\eta}))$ from the $L_{\eta\eta}$ smoothness of $A$ in $\eta$ gives

$$\|\nabla_\eta A(\hat\eta, \theta)\|^2 \leq L_{\eta\eta}^2 \|\eta^{0,0} - \eta^*\|^2 \left(1 - \frac{z}{L_{\eta\eta}}\right)^{MN+1} + \frac{2G^2}{L_{\eta\eta}} [\mathcal{M}_K N - 1]^2 \left(\frac{2}{z} + \frac{1}{2L_{\eta\eta}}\right)$$

$$\leq D(\mathcal{X}) L_{\eta\eta}^2 \left(1 - \frac{z}{L_{\eta\eta}}\right)^{MN+1} + \frac{2G^2}{L_{\eta\eta}} [\mathcal{M}_K N - 1]^2 \left(\frac{2}{z} + \frac{1}{2L_{\eta\eta}}\right) \quad \text{(B.40)}$$

Where $\eta^*$ is the true solution to the inner maximization problem. Since we initialize $\eta^{0,0} \in \mathcal{X}$, we have that $\|\eta^{0,0} - \eta^*\|^2 \leq D(\mathcal{X})$. We can get:

$$\mathbb{E}\|\nabla_\eta x_{t_c,i}\|^2 \leq D(\mathcal{X}) L_{\eta\eta}^2 \left(1 - \frac{z}{L_{\eta\eta}}\right)^{MN+1} + \frac{2G^2}{L_{\eta\eta}} [\mathcal{M}_K N - 1]^2 \left(\frac{2}{z} + \frac{1}{2L_{\eta\eta}}\right) \quad \text{(B.41)}$$

For (b): we use **Lemma 3**, and **Assumption 1**:

$$\mathbb{E}\|\hat\nabla_{x_{t_c}} \hat{\mathcal{A}}_i(\hat\eta_i, \psi_i, \theta_i)\|^2$$

$$\leq 3\mathbb{E}\|\hat\nabla_{x_{t_c}} \hat{\mathcal{A}}_i(\hat\eta_i, \psi_i, \theta_i) - \hat\nabla_{x_{t_c}} \mathcal{A}_i(\hat\eta_i, \psi_i, \theta_i)\|^2 + 3\mathbb{E}\|\hat\nabla_{x_{t_c}} \mathcal{A}_i(\hat\eta_i, \psi_i, \theta_i) - \nabla_{x_{t_c}} \mathcal{A}_i(\hat\eta_i, \psi_i, \theta_i)\|^2$$

$$+ 3\mathbb{E}\|\nabla_{x_{t_c}} \mathcal{A}_i(\hat\eta_i, \psi_i, \theta_i)\|^2$$

$$\leq 3\left(H_\theta^2 C \mathcal{E}^k + \frac{L^2 \mu^2 d^2}{4} + K^2\right) \quad \text{(B.42)}$$

Substituting (a) and (b) completes the proof.

**Lemma 7.** *Connecting Gradients.* *Under the **Assumption 1**, **Assumption 5**, and **Lemma 2**, the following inequality can be obtained:*

$$\mathbb{E}\|\nabla_\psi \mathcal{G}_\mathcal{B}(\hat\eta, \psi, \theta) - \nabla_\psi \mathcal{G}_\mathcal{B}(\eta^*, \psi, \theta)\|^2 \leq \frac{L_{\psi\eta}^2 \cdot \zeta^k}{z^2} \quad \text{(B.43)}$$

$$\mathbb{E}\|\nabla_\theta \mathcal{G}_\mathcal{B}(\hat\eta, \psi, \theta) - \nabla_\theta \mathcal{G}_\mathcal{B}(\eta^*, \psi, \theta)\|^2 \leq \frac{L_{\theta\eta}^2 \cdot \zeta^k}{z^2} \quad \text{(B.44)}$$

*where* $\zeta^k = 3\left(H_\theta^2 C \mathcal{E}^k + \frac{L^2 \mu^2 d^2}{4} + K^2\right)\pi(M, N)$,
$\pi(M, N) = \left\{ D(\mathcal{X}) L_{\eta\eta}^2 \left(1 - \frac{z}{L_{\eta\eta}}\right)^{MN+1} + \frac{2G^2}{L_{\eta\eta}} [\mathcal{M}_K N - 1]^2 \left(\frac{2}{z} + \frac{1}{2L_{\eta\eta}}\right)\right\}$.

**Proof:**
Under **Assumption 1**, **Assumption 5**, and **Lemma 2**, for server model parameters $\psi$, we can get:

$$\mathbb{E}\|\nabla_\psi \mathcal{G}_\mathcal{B}(\hat\eta, \psi, \theta) - \nabla_\psi \mathcal{G}_\mathcal{B}(\eta^*, \psi, \theta)\|^2 \leq \frac{1}{B}\sum_{i\in\mathcal{B}} \mathbb{E}\|\nabla_\psi \mathcal{A}_i(\hat\eta_i, \psi_i, \theta_i) - \nabla_\psi \mathcal{A}_i(\eta_i^*, \psi_i, \theta_i)\|^2$$

$$\leq \frac{L_{\psi\eta}^2}{B}\sum_{i\in\mathcal{B}} \mathbb{E}\|\hat\eta_i - \eta_i^*\|^2$$

$$\leq \frac{L_{\psi\eta}^2}{Bz^2}\sum_{i\in\mathcal{B}} \mathbb{E}\|\nabla_\eta \mathcal{A}_i(\hat\eta_i, \psi_i, \theta_i)\|^2$$

$$\leq \frac{L_{\psi\eta}^2 \cdot \zeta^k}{z^2} \quad \text{(B.45)}$$

Similar to the proof for $\psi$ in (B.45), for client model parameters $\theta$, we get:

$$\mathbb{E}\|\nabla_\theta \mathcal{G}_\mathcal{B}(\hat\eta, \psi, \theta) - \nabla_\theta \mathcal{G}_\mathcal{B}(\eta^*, \psi, \theta)\|^2 \leq \frac{L_{\theta\eta}^2 \cdot \zeta^k}{z^2} \quad \text{(B.46)}$$

**Theorem 1.** *Bound the Global Update Round.* *When the parameters are updated with the perturbations:*

$$\hat\eta_i = \underset{\substack{m=1,\ldots,M \\ n=1,\ldots,N}}{argmin} \left\|\hat\nabla_\eta \hat{\mathcal{A}}_i(\eta_i^{m,n}, \psi_i, \theta_i)\right\| \quad \text{(B.47)}$$

*The gradient of $\hat{\eta}_i$ is bounded:*

$$\mathbb{E}||\hat{\nabla}_\eta \hat{\mathcal{A}}_i(\hat{\eta}_i, \psi_i, \theta_i)||^2 \leq \zeta^k \tag{B.48}$$

*where* $\zeta^k = 3(H_\theta^2 C \mathcal{E}^k + \frac{L^2 \mu^2 d^2}{4} + K^2)\pi(M, N)$,
$\pi(M, N) = \left\{ D(\mathcal{X})L_{\eta\eta}^2 \left(1 - \frac{z}{L_{\eta\eta}}\right)^{MN+1} + \frac{2G^2}{L_{\eta\eta}}\left[\mathcal{M}_K N - 1\right]^2 \left(\frac{2}{z} + \frac{1}{2L_{\eta\eta}}\right) \right\}$.

*The global iterates satisfy:*

$$\frac{1}{\mathcal{K}} \sum_{k=0}^{\mathcal{K}-1} \mathbb{E}\left[||\nabla \mathcal{R}\left(\eta^{*,k}, \psi^k, \theta^k\right)||^2\right]$$

$$\leq \left(\frac{2\Lambda}{\alpha_m \mathcal{K}}\right) + \left(\frac{2L_\psi \alpha_M^2 \sigma_\psi^2}{\alpha_m} + \frac{2L_\theta \alpha_M^2 \sigma_\theta^2}{\alpha_m}\right)$$

$$+ \mathcal{E}\frac{\alpha_M}{\alpha_m}\left[2(1 + L_\psi \alpha_M)H_\psi^2 C + 3(1 + L_\theta \alpha_M)Q_\theta^2 H_\theta^2 C + \frac{3H_\theta^2 C \pi(M, N)}{z^2}(2(1 + L_\psi \alpha_M)L_{\psi\eta}^2 + 3(1 + L_\theta \alpha_M)L_{\theta\eta}^2)\right]$$

$$+ \mu^2\left(\frac{3\alpha_M(1 + L_\theta \alpha_M)L^2 d^2}{4\alpha_m} + \frac{3\pi(M, N)L^2 d^2 a_M(1 + L_\psi \alpha_M)L_{\psi\eta}^2}{2a_m z^2} + \frac{9\pi(M, N)L^2 d^2 a_M(1 + L_\theta \alpha_M)L_{\theta\eta}^2}{4a_m z^2}\right)$$

$$+ \frac{3\alpha_M K^2 \pi(M, N)}{\alpha_m z^2}\left[2(1 + L_\psi \alpha_M)L_{\psi\eta}^2 + 3(1 + L_\theta \alpha_M)L_{\theta\eta}^2\right] \tag{B.49}$$

**Proof:**

For the gradient respect to $\psi$, there exists compression error, but no estimation error: $\nabla_\psi \mathcal{G}_\mathcal{B}(\hat{\eta}^k, \psi^k, \theta^k) := (1/B)\sum_{i \in \mathcal{B}} \nabla_\psi \hat{\mathcal{A}}_i(\hat{\eta}_i^k, \psi_i^k, \theta_i^k)$, where $\hat{\eta}_i^k$ is the output of the adversary's inner problem at iteration $k$, $\hat{\eta}_i^k$ and $\nabla_\psi \widehat{\mathcal{G}}_\mathcal{B}(\hat{\eta}^k, \psi^k, \theta^k)$ satisfy the following equations:

$$\hat{\eta}_i = \underset{\substack{m=1,...,M \\ n=1,...,N}}{argmin} \left\| \hat{\nabla}_\eta \hat{\mathcal{A}}_i\left(\eta_i^{m,n}, \psi_i, \theta_i\right) \right\| \tag{B.50}$$

$$\psi^{k+1} = \psi^k - \alpha_\psi \cdot \nabla_\psi \widehat{\mathcal{G}}_\mathcal{B}(\hat{\eta}^k, \psi^k, \theta^k) \tag{B.51}$$

For the gradient respect to $\theta$, there exist compression error and estimation error: $\hat{\nabla}_\theta \hat{\mathcal{G}}_\mathcal{B}(\hat{\eta}^k, \psi^k, \theta^k) := (1/B)\sum_{i \in \mathcal{B}} \hat{\nabla}_\theta \hat{\mathcal{A}}_i(\hat{\eta}_i^k, \psi_i^k, \theta_i^k)$, the $\hat{\nabla}_\theta \hat{\mathcal{G}}_\mathcal{B}(\hat{\eta}^k, \psi^k, \theta^k)$ satisfy the following equation:

$$\theta^{k+1} = \theta^k - \alpha_\theta \cdot \hat{\nabla}_\theta \hat{\mathcal{G}}_\mathcal{B}(\hat{\eta}^k, \psi^k, \theta^k) \tag{B.52}$$

Furthermore, $\nabla_\psi \mathcal{G}_\mathcal{B}(\eta^{*,k}, \psi^k, \theta^k)$ and $\nabla_\theta \mathcal{G}_\mathcal{B}(\eta^{*,k}, \psi^k, \theta^k)$ are true stochastic gradients, $\nabla_\psi \mathcal{R}(\eta^{*,k}, \psi^k, \theta^k)$ and $\nabla_\theta \mathcal{R}(\eta^{*,k}, \psi^k, \theta^k)$ are true full gradients.
We begin with the inequality for the $L$-smoothness of $\nabla \mathcal{R}(\eta^{*,k}, \psi^k, \theta^k)$, and apply **Proposition1**, $k \in 0, 1, ..., \mathcal{K}$ is the iteration indice, we can get:

$$\mathcal{R}\left(\eta^{*,k+1}, \psi^{k+1}, \theta^{k+1}\right) - \mathcal{R}\left(\eta^{*,k}, \psi^k, \theta^k\right)$$

$$\leq \underbrace{\left\langle \nabla_\psi \mathcal{R}\left(\eta^{*,k}, \psi^k, \theta^k\right), \psi^{k+1} - \psi^k\right\rangle + \frac{L_\psi}{2}\left\|\psi^{k+1} - \psi^k\right\|^2}_{a}$$

$$+ \underbrace{\left\langle \nabla_\theta \mathcal{R}\left(\eta^{*,k}, \psi^k, \theta^k\right), \theta^{k+1} - \theta^k\right\rangle + \frac{L_\theta}{2}\left\|\theta^{k+1} - \theta^k\right\|^2}_{b} \tag{B.53}$$

For (a):

$$\left\langle \nabla_\psi \mathcal{R}\left(\eta^{*,k}, \psi^k, \theta^k\right), \psi^{k+1} - \psi^k \right\rangle + \frac{L_\psi}{2} \left\| \psi^{k+1} - \psi^k \right\|^2$$

$$= \left\langle \nabla_\psi \mathcal{R}\left(\eta^{*,k}, \psi^k, \theta^k\right), -\alpha_\psi \cdot \nabla_\psi \widehat{\mathcal{G}}_\mathcal{B}(\hat\eta^k, \psi^k, \theta^k) \right\rangle + \frac{L_\psi \alpha_\psi^2}{2} \left\| \nabla_\psi \widehat{\mathcal{G}}_\mathcal{B}(\hat\eta^k, \psi^k, \theta^k) \right\|^2$$

$$= -\alpha_\psi(1 - \frac{L_\psi \alpha_\psi}{2})||\nabla_\psi \mathcal{R}\left(\eta^{*,k}, \psi^k, \theta^k\right)||^2 + \frac{L_\psi \alpha_\psi^2}{2}||\nabla_\psi \widehat{\mathcal{G}}_\mathcal{B}(\hat\eta^k, \psi^k, \theta^k) - \nabla_\psi \mathcal{R}\left(\eta^{*,k}, \psi^k, \theta^k\right)||^2$$
$$+ \alpha_\psi(1 - L_\psi \alpha_\psi) \left\langle \nabla_\psi \mathcal{R}\left(\eta^{*,k}, \psi^k, \theta^k\right), \nabla_\psi \mathcal{R}\left(\eta^{*,k}, \psi^k, \theta^k\right) - \nabla_\psi \widehat{\mathcal{G}}_\mathcal{B}(\hat\eta^k, \psi^k, \theta^k) \right\rangle$$

$$= -\alpha_\psi(1 - \frac{L_\psi \alpha_\psi}{2})||\nabla_\psi \mathcal{R}\left(\eta^{*,k}, \psi^k, \theta^k\right)||^2 + \frac{L_\psi \alpha_\psi^2}{2}||\nabla_\psi \widehat{\mathcal{G}}_\mathcal{B}(\hat\eta^k, \psi^k, \theta^k) - \nabla_\psi \mathcal{R}\left(\eta^{*,k}, \psi^k, \theta^k\right)||^2$$
$$+ \alpha_\psi(1 - L_\psi \alpha_\psi) \left\langle \nabla_\psi \mathcal{R}\left(\eta^{*,k}, \psi^k, \theta^k\right), \nabla_\psi \mathcal{R}\left(\eta^{*,k}, \psi^k, \theta^k\right) - \nabla_\psi \mathcal{G}_\mathcal{B}(\eta^{*,k}, \psi^k, \theta^k) \right\rangle$$
$$+ \alpha_\psi(1 - L_\psi \alpha_\psi) \left\langle \nabla_\psi \mathcal{R}\left(\eta^{*,k}, \psi^k, \theta^k\right), \nabla_\psi \mathcal{G}_\mathcal{B}(\eta^{*,k}, \psi^k, \theta^k) - \nabla_\psi \widehat{\mathcal{G}}_\mathcal{B}(\hat\eta^k, \psi^k, \theta^k) \right\rangle$$

$$\leq -\alpha_\psi(1 - \frac{L_\psi \alpha_\psi}{2})||\nabla_\psi \mathcal{R}\left(\eta^{*,k}, \psi^k, \theta^k\right)||^2 + L_\psi \alpha_\psi^2||\nabla_\psi \widehat{\mathcal{G}}_\mathcal{B}(\hat\eta^k, \psi^k, \theta^k) - \nabla_\psi \mathcal{G}_\mathcal{B}(\eta^{*,k}, \psi^k, \theta^k)||^2$$
$$+ L_\psi \alpha_\psi^2||\nabla_\psi \mathcal{G}_\mathcal{B}(\eta^{*,k}, \psi^k, \theta^k) - \nabla_\psi \mathcal{R}\left(\eta^{*,k}, \psi^k, \theta^k\right)||^2$$
$$+ \alpha_\psi(1 - L_\psi \alpha_\psi) \left\langle \nabla_\psi \mathcal{R}\left(\eta^{*,k}, \psi^k, \theta^k\right), \nabla_\psi \mathcal{R}\left(\eta^{*,k}, \psi^k, \theta^k\right) - \nabla_\psi \mathcal{G}_\mathcal{B}(\eta^{*,k}, \psi^k, \theta^k) \right\rangle$$
$$+ \frac{\alpha_\psi}{2}(1 - L_\psi \alpha_\psi)||\nabla_\psi \mathcal{R}\left(\eta^{*,k}, \psi^k, \theta^k\right)||^2 + \frac{\alpha_\psi}{2}(1 - L_\psi \alpha_\psi)||\nabla_\psi \mathcal{G}_\mathcal{B}(\eta^{*,k}, \psi^k, \theta^k) - \nabla_\psi \widehat{\mathcal{G}}_\mathcal{B}(\hat\eta^k, \psi^k, \theta^k)||^2$$

$$= -\frac{\alpha_\psi}{2}||\nabla_\psi \mathcal{R}\left(\eta^{*,k}, \psi^k, \theta^k\right)||^2 + \frac{\alpha_\psi}{2}(1 + L_\psi \alpha_\psi)||\nabla_\psi \widehat{\mathcal{G}}_\mathcal{B}(\hat\eta^k, \psi^k, \theta^k) - \nabla_\psi \mathcal{G}_\mathcal{B}(\eta^{*,k}, \psi^k, \theta^k)||^2$$
$$+ L_\psi \alpha_\psi^2||\nabla_\psi \mathcal{G}_\mathcal{B}(\eta^{*,k}, \psi^k, \theta^k) - \nabla_\psi \mathcal{R}\left(\eta^{*,k}, \psi^k, \theta^k\right)||^2$$
$$+ \alpha_\psi(1 - L_\psi \alpha_\psi) \left\langle \nabla_\psi \mathcal{R}\left(\eta^{*,k}, \psi^k, \theta^k\right), \nabla_\psi \mathcal{R}\left(\eta^{*,k}, \psi^k, \theta^k\right) - \nabla_\psi \mathcal{G}_\mathcal{B}(\eta^{*,k}, \psi^k, \theta^k) \right\rangle \quad \text{(B.54)}$$

Note that $\mathbb{E}\left[\nabla_\psi \mathcal{G}_\mathcal{B}(\eta^{*,k}, \psi^k, \theta^k)\right] = \nabla_\psi \mathcal{R}\left(\eta^{*,k}, \psi^k, \theta^k\right)$, where the expectation is taken over the randomness of the mini-batch sampling. We can get:

$$\mathbb{E}\left[\nabla_\psi \mathcal{R}\left(\eta^{*,k}, \psi^k, \theta^k\right), \nabla_\psi \mathcal{R}\left(\eta^{*,k}, \psi^k, \theta^k\right) - \nabla_\psi \mathcal{G}_\mathcal{B}(\eta^{*,k}, \psi^k, \theta^k)\right] = 0 \quad \text{(B.55)}$$

Then, we can get:

$$\mathbb{E}\left[\left\langle \nabla_\psi \mathcal{R}\left(\eta^{*,k}, \psi^k, \theta^k\right), \psi^{k+1} - \psi^k \right\rangle + \frac{L_\psi}{2} \left\| \psi^{k+1} - \psi^k \right\|^2\right]$$

$$\leq -\frac{\alpha_\psi}{2}\mathbb{E}\left[||\nabla_\psi \mathcal{R}\left(\eta^{*,k}, \psi^k, \theta^k\right)||^2\right] + \frac{\alpha_\psi}{2}(1 + L_\psi \alpha_\psi)\mathbb{E}\left[||\nabla_\psi \widehat{\mathcal{G}}_\mathcal{B}(\hat\eta^k, \psi^k, \theta^k) - \nabla_\psi \mathcal{G}_\mathcal{B}(\eta^{*,k}, \psi^k, \theta^k)||^2\right]$$
$$+ L_\psi \alpha_\psi^2 \mathbb{E}\left[||\nabla_\psi \mathcal{G}_\mathcal{B}(\eta^{*,k}, \psi^k, \theta^k) - \nabla_\psi \mathcal{R}\left(\eta^{*,k}, \psi^k, \theta^k\right)||^2\right] \quad \text{(B.56)}$$

Under **Assumption 2**, we can get:

$$\mathbb{E}\left[||\nabla_\psi \mathcal{G}_\mathcal{B}(\eta^{*,k}, \psi^k, \theta^k) - \nabla_\psi \mathcal{R}\left(\eta^{*,k}, \psi^k, \theta^k\right)||^2\right] \leq \sigma_\psi^2 \quad \text{(B.57)}$$

Furthermore, under **Lemma 5** and **Lemma 7**, we can get:

$$\mathbb{E}\left[||\nabla_\psi \widehat{\mathcal{G}}_\mathcal{B}(\hat\eta^k, \psi^k, \theta^k) - \nabla_\psi \mathcal{G}_\mathcal{B}(\eta^{*,k}, \psi^k, \theta^k)||^2\right]$$

$$\leq 2\mathbb{E}\left[||\nabla_\psi \widehat{\mathcal{G}}_\mathcal{B}(\hat\eta^k, \psi^k, \theta^k) - \nabla_\psi \mathcal{G}_\mathcal{B}(\hat\eta^k, \psi^k, \theta^k)||^2\right] + 2\mathbb{E}\left[||\nabla_\psi \mathcal{G}_\mathcal{B}(\hat\eta^k, \psi^k, \theta^k) - \nabla_\psi \mathcal{G}_\mathcal{B}(\eta^{*,k}, \psi^k, \theta^k)||^2\right]$$

$$\leq 2H_\psi^2 C \mathcal{E}^k + 2\frac{L_{\psi\eta}^2 \cdot \zeta^k}{z^2} \quad \text{(B.58)}$$

Finally, we can be obtained:

$$\mathbb{E}\left[\left\langle \nabla_\psi \mathcal{R}\left(\eta^{*,k}, \psi^k, \theta^k\right), \psi^{k+1} - \psi^k \right\rangle + \frac{L_\psi}{2} \left\| \psi^{k+1} - \psi^k \right\|^2\right]$$

$$\leq -\frac{\alpha_\psi}{2}\mathbb{E}\left[||\nabla_\psi \mathcal{R}\left(\eta^{*,k}, \psi^k, \theta^k\right)||^2\right] + \frac{\alpha_\psi}{2}(1 + L_\psi \alpha_\psi)(2H_\psi^2 V \mathcal{E}^k + 2\frac{L_{\psi\eta}^2 \cdot \zeta^k}{z^2}) + L_\psi \alpha_\psi^2 \sigma_\psi^2$$
$$\text{(B.59)}$$

For (b), similar to the proof for $\psi$ in **B.56**), for $\theta$, we can get:

$$\mathbb{E}\left[\left\langle\nabla_\theta\mathcal{R}\left(\eta^{*,k},\psi^k,\theta^k\right),\theta^{k+1}-\theta^k\right\rangle+\frac{L_\psi}{2}\left\|\theta^{k+1}-\theta^k\right\|^2\right]$$

$$\leq-\frac{\alpha_\theta}{2}\mathbb{E}\left[||\nabla_\theta\mathcal{R}\left(\eta^{*,k},\psi^k,\theta^k\right)||^2\right]+\frac{\alpha_\theta}{2}(1+L_\theta\alpha_\theta)\mathbb{E}\left[||\hat{\nabla}_\theta\hat{\mathcal{G}}_\mathcal{B}(\hat{\eta}^k,\psi^k,\theta^k)-\nabla_\theta\mathcal{G}_\mathcal{B}(\eta^{*,k},\psi^k,\theta^k)||^2\right]$$

$$+L_\theta\alpha_\theta^2\mathbb{E}\left[||\nabla_\theta\mathcal{G}_\mathcal{B}(\eta^{*,k},\psi^k,\theta^k)-\nabla_\theta\mathcal{R}\left(\eta^{*,k},\psi^k,\theta^k\right)||^2\right]$$

$$+\alpha_\theta(1-L_\theta\alpha_\theta)\mathbb{E}\left[\nabla_\theta\mathcal{R}\left(\eta^{*,k},\psi^k,\theta^k\right),\nabla_\theta\mathcal{R}\left(\eta^{*,k},\psi^k,\theta^k\right)-\nabla_\theta\mathcal{G}_\mathcal{B}(\eta^{*,k},\psi^k,\theta^k)\right]\quad\text{(B.60)}$$

Under **Assumption 2**, we can get:

$$\mathbb{E}\left[nabla_\theta\mathcal{R}\left(\eta^{*,k},\psi^k,\theta^k\right),\nabla_\theta\mathcal{R}\left(\eta^{*,k},\psi^k,\theta^k\right)-\nabla_\theta\mathcal{G}_\mathcal{B}(\eta^{*,k},\psi^k,\theta^k)\right]=0\quad\text{(B.61)}$$

$$\mathbb{E}\left[||\nabla_\theta\mathcal{G}_\mathcal{B}(\eta^{*,k},\psi^k,\theta^k)-\nabla_\theta\mathcal{R}\left(\eta^{*,k},\psi^k,\theta^k\right)||^2\right]\leq\sigma_\theta^2\quad\text{(B.62)}$$

Furthermore, under **Lemma 3**, **Lemma 5** and **Lemma 7**, we can get:

$$\mathbb{E}\left[||\hat{\nabla}_\theta\hat{\mathcal{G}}_\mathcal{B}(\hat{\eta}^k,\psi^k,\theta^k)-\nabla_\theta\mathcal{G}_\mathcal{B}(\eta^{*,k},\psi^k,\theta^k)||^2\right]$$

$$\leq3\mathbb{E}\left[||\hat{\nabla}_\theta\hat{\mathcal{G}}_\mathcal{B}(\hat{\eta}^k,\psi^k,\theta^k)-\nabla_\theta\hat{\mathcal{G}}_\mathcal{B}(\hat{\eta}^k,\psi^k,\theta^k)||^2\right]+3\mathbb{E}\left[||\nabla_\theta\hat{\mathcal{G}}_\mathcal{B}(\hat{\eta}^k,\psi^k,\theta^k)-\nabla_\theta\mathcal{G}_\mathcal{B}(\hat{\eta}^k,\psi^k,\theta^k)||^2\right]$$

$$+3\mathbb{E}\left[||\nabla_\theta\mathcal{G}_\mathcal{B}(\hat{\eta}^k,\psi^k,\theta^k)-\nabla_\theta\mathcal{G}_\mathcal{B}(\eta^{*,k},\psi^k,\theta^k)||^2\right]$$

$$\leq3\frac{L^2\mu^2d^2}{4}+3Q_\theta^2H_\theta^2C\mathcal{E}^k+3\frac{L_{\theta\eta}^2\cdot\zeta^k}{z^2}\quad\text{(B.63)}$$

Finally, we can be obtained:

$$\mathbb{E}\left[\left\langle\nabla_\psi\mathcal{R}\left(\eta^{*,k},\psi^k,\theta^k\right),\psi^{k+1}-\psi^k\right\rangle+\frac{L_\psi}{2}\left\|\psi^{k+1}-\psi^k\right\|^2\right]$$

$$\leq-\frac{\alpha_\theta}{2}\mathbb{E}\left[||\nabla_\theta\mathcal{R}\left(\eta^{*,k},\psi^k,\theta^k\right)||^2\right]+\frac{\alpha_\theta}{2}(1+L_\theta\alpha_\theta)(3\frac{L^2\mu^2d^2}{4}+3Q_\theta^2H_\theta^2C\mathcal{E}^k+3\frac{L_{\theta\eta}^2\cdot\zeta^k}{z^2})+L_\theta\alpha_\theta^2\sigma_\theta^2$$
$$\text{(B.64)}$$

Substituting a) and b), we can get:

$$\mathbb{E}\left[\mathcal{R}\left(\eta^{*,k+1},\psi^{k+1},\theta^{k+1}\right)-\mathcal{R}\left(\eta^{*,k},\psi^k,\theta^k\right)\right]$$

$$\leq\mathbb{E}\left[\left\langle\nabla_\psi\mathcal{R}\left(\eta^{*,k},\psi^k,\theta^k\right),\psi^{k+1}-\psi^k\right\rangle+\frac{L_\psi}{2}\left\|\psi^{k+1}-\psi^k\right\|^2\right]$$

$$+\mathbb{E}\left[\left\langle\nabla_\theta\mathcal{R}\left(\eta^{*,k},\psi^k,\theta^k\right),\theta^{k+1}-\theta^k\right\rangle+\frac{L_\theta}{2}\left\|\theta^{k+1}-\theta^k\right\|^2\right]$$

$$\leq-\frac{\alpha_\psi}{2}\mathbb{E}\left[||\nabla_\psi\mathcal{R}\left(\eta^{*,k},\psi^k,\theta^k\right)||^2\right]+\frac{\alpha_\psi}{2}(1+L_\psi\alpha_\psi)(2H_\psi^2C\mathcal{E}^k+2\frac{L_{\psi\eta}^2\cdot\zeta^k}{z^2})+L_\psi\alpha_\psi^2\sigma_\psi^2$$

$$-\frac{\alpha_\theta}{2}\mathbb{E}\left[||\nabla_\theta\mathcal{R}\left(\eta^{*,k},\psi^k,\theta^k\right)||^2\right]+\frac{\alpha_\theta}{2}(1+L_\theta\alpha_\theta)(3\frac{L^2\mu^2d^2}{4}+3Q_\theta^2H_\theta^2C\mathcal{E}^k+3\frac{L_{\theta\eta}^2\cdot\zeta^k}{z^2})+L_\theta\alpha_\theta^2\sigma_\theta^2$$
$$\text{(B.65)}$$

Since $\psi$ and $\theta$ are updated synchronously in the outer loop, we take $\alpha_m=\min\{\alpha_\psi,\alpha_\theta\}$, and combine the gradient:

$$\mathbb{E}\left[\mathcal{R}\left(\eta^{*,k+1},\psi^{k+1},\theta^{k+1}\right)-\mathcal{R}\left(\eta^{*,k},\psi^k,\theta^k\right)\right]$$

$$\leq-\frac{\alpha_m}{2}\mathbb{E}\left[||\nabla\mathcal{R}\left(\eta^{*,k},\psi^k,\theta^k\right)||^2\right]+\frac{\alpha_\psi}{2}(1+L_\psi\alpha_\psi)(2H_\psi^2C\mathcal{E}^k+2\frac{L_{\psi\eta}^2\cdot\zeta^k}{z^2})+L_\psi\alpha_\psi^2\sigma_\psi^2$$

$$+\frac{\alpha_\theta}{2}(1+L_\theta\alpha_\theta)(3\frac{L^2\mu^2d^2}{4}+3Q_\theta^2H_\theta^2C\mathcal{E}^k+3\frac{L_{\theta\eta}^2\cdot\zeta^k}{z^2})+L_\theta\alpha_\theta^2\sigma_\theta^2\quad\text{(B.66)}$$

Summing these inequalities from $k = 0$ to $\mathcal{K} - 1$, take $\mathcal{E} = \max\limits_{k=0,\ldots,K-1}(\mathcal{E}^k)$, and then $\zeta = \max\limits_{k=0,\ldots,K-1}(\zeta^k)$:

$$\frac{1}{\mathcal{K}} \sum_{k=0}^{\mathcal{K}-1} \frac{\alpha_m}{2} \mathbb{E}\left[||\nabla \mathcal{R}\left(\eta^{*,k}, \psi^k, \theta^k\right)||^2\right]$$

$$\leq \frac{1}{\mathcal{K}} \sum_{k=0}^{\mathcal{K}-1} \mathbb{E}\left[\mathcal{R}\left(\eta^{*,k}, \psi^k, \theta^k\right) - \mathcal{R}\left(\eta^{*,k+1}, \psi^{k+1}, \theta^{k+1}\right)\right] + \frac{\alpha_\psi}{2}(1 + L_\psi \alpha_\psi)(2H_\psi^2 C\mathcal{E} + 2\frac{L_{\psi\eta}^2 \cdot \zeta}{z^2}) + L_\psi \alpha_\psi^2 \sigma_\psi^2$$

$$+ \frac{\alpha_\theta}{2}(1 + L_\theta \alpha_\theta)(3\frac{L^2 \mu^2 d^2}{4} + 3Q_\theta^2 H_\theta^2 C\mathcal{E} + 3\frac{L_{\theta\eta}^2 \cdot \zeta}{z^2}) + L_\theta \alpha_\theta^2 \sigma_\theta^2$$

$$= \mathbb{E}\left[\mathcal{R}\left(\eta^{*,0}, \psi^0, \theta^0\right) - \mathcal{R}\left(\eta^{*,\mathcal{K}}, \psi^\mathcal{K}, \theta^\mathcal{K}\right)\right] + \frac{\alpha_\psi}{2}(1 + L_\psi \alpha_\psi)(2H_\psi^2 C\mathcal{E} + 2\frac{L_{\psi\eta}^2 \cdot \zeta}{z^2}) + L_\psi \alpha_\psi^2 \sigma_\psi^2$$

$$+ \frac{\alpha_\theta}{2}(1 + L_\theta \alpha_\theta)(3\frac{L^2 \mu^2 d^2}{4} + 3Q_\theta^2 H_\theta^2 C\mathcal{E} + 3\frac{L_{\theta\eta}^2 \cdot \zeta}{z^2}) + L_\theta \alpha_\theta^2 \sigma_\theta^2 \tag{B.67}$$

Then, we define $\Lambda = \mathcal{R}\left(\eta^{*,0}, \psi^0, \theta^0\right) - \inf_k(\mathcal{R}\left(\eta^{*,k}, \psi^k, \theta^k\right))$ and $\alpha_M = \max\{\alpha_\psi, \alpha_\theta\}$:

$$\frac{1}{\mathcal{K}} \sum_{k=0}^{\mathcal{K}-1} \mathbb{E}\left[||\nabla \mathcal{R}\left(\eta^{*,k}, \psi^k, \theta^k\right)||^2\right]$$

$$\leq \frac{2\Lambda}{\alpha_m \mathcal{K}} + \frac{\alpha_\psi(1 + L_\psi \alpha_\psi)}{\alpha_m}(2H_\psi^2 C\mathcal{E} + 2\frac{L_{\psi\eta}^2 \cdot \zeta}{z^2}) + \frac{2L_\psi \alpha_\psi^2 \sigma_\psi^2}{\alpha_m}$$

$$+ \frac{\alpha_\theta(1 + L_\theta \alpha_\theta)}{\alpha_m}(3\frac{L^2 \mu^2 d^2}{4} + 3Q_\theta^2 H_\theta^2 C\mathcal{E} + 3\frac{L_{\theta\eta}^2 \cdot \zeta}{z^2}) + \frac{2L_\theta \alpha_\theta^2 \sigma_\theta^2}{\alpha_m}$$

$$\leq \frac{2\Lambda}{\alpha_m \mathcal{K}} + \frac{\alpha_M(1 + L_\psi \alpha_M)}{\alpha_m}(2H_\psi^2 C\mathcal{E} + 2\frac{L_{\psi\eta}^2 \cdot \zeta}{z^2}) + \frac{2L_\psi \alpha_M^2 \sigma_\psi^2}{\alpha_m}$$

$$+ \frac{\alpha_M(1 + L_\theta \alpha_M)}{\alpha_m}(3\frac{L^2 \mu^2 d^2}{4} + 3Q_\theta^2 H_\theta^2 C\mathcal{E} + 3\frac{L_{\theta\eta}^2 \cdot \zeta}{z^2}) + \frac{2L_\theta \alpha_M^2 \sigma_\theta^2}{\alpha_m}$$

$$= (\frac{2\Lambda}{\alpha_m \mathcal{K}} + \frac{2L_\psi \alpha_M^2 \sigma_\psi^2}{\alpha_m} + \frac{2L_\theta \alpha_M^2 \sigma_\theta^2}{\alpha_m}) + \frac{\alpha_M}{\alpha_m}\left[2(1 + L_\psi \alpha_M)H_\psi^2 C\mathcal{E} + 3(1 + L_\theta \alpha_M)Q_\theta^2 H_\theta^2 C\mathcal{E}\right]$$

$$+ \frac{3\alpha_M(1 + L_\theta \alpha_M)L^2 \mu^2 d^2}{4\alpha_m} + \frac{\alpha_M \tau}{\alpha_m z^2}\left[2(1 + L_\psi \alpha_M)L_{\psi\eta}^2 + 3(1 + L_\theta \alpha_M)L_{\theta\eta}^2\right]$$

$$= (\frac{2\Lambda}{\alpha_m \mathcal{K}} + \frac{2L_\psi \alpha_M^2 \sigma_\psi^2}{\alpha_m} + \frac{2L_\theta \alpha_M^2 \sigma_\theta^2}{\alpha_m}) + \mathcal{E}\frac{\alpha_M}{\alpha_m}\left[2(1 + L_\psi \alpha_M)H_\psi^2 C + 3(1 + L_\theta \alpha_M)Q_\theta^2 H_\theta^2 C\right]$$

$$+ \mu^2 \frac{3\alpha_M(1 + L_\theta \alpha_M)L^2 d^2}{4\alpha_m} + \frac{\alpha_M \tau}{\alpha_m z^2}\left[2(1 + L_\psi \alpha_M)L_{\psi\eta}^2 + 3(1 + L_\theta \alpha_M)L_{\theta\eta}^2\right]$$

$$= (\frac{2\Lambda}{\alpha_m \mathcal{K}}) + (\frac{2L_\psi \alpha_M^2 \sigma_\psi^2}{\alpha_m} + \frac{2L_\theta \alpha_M^2 \sigma_\theta^2}{\alpha_m})$$

$$+ \mathcal{E}\frac{\alpha_M}{\alpha_m}\left[2(1 + L_\psi \alpha_M)H_\psi^2 C + 3(1 + L_\theta \alpha_M)Q_\theta^2 H_\theta^2 C + \frac{3H_\theta^2 C\pi(M,N)}{z^2}(2(1 + L_\psi \alpha_M)L_{\psi\eta}^2 + 3(1 + L_\theta \alpha_M)L_{\theta\eta}^2)\right]$$

$$+ \mu^2(\frac{3\alpha_M(1 + L_\theta \alpha_M)L^2 d^2}{4\alpha_m} + \frac{3\pi(M,N)L^2 d^2 a_M(1 + L_\psi \alpha_M)L_{\psi\eta}^2}{2a_m z^2} + \frac{9\pi(M,N)L^2 d^2 a_M(1 + L_\theta \alpha_M)L_{\theta\eta}^2}{4a_m z^2})$$

$$+ \frac{3\alpha_M K^2 \pi(M,N)}{\alpha_m z^2}\left[2(1 + L_\psi \alpha_M)L_{\psi\eta}^2 + 3(1 + L_\theta \alpha_M)L_{\theta\eta}^2\right] \tag{B.68}$$

**Corollary 1** According to **Theorem 1**:If we choose $\alpha_\theta$ and $\alpha_\psi$ as $\mathcal{O}(\frac{1}{\sqrt{\mathcal{K}}}), \mu = \mathcal{O}(\frac{1}{\mathcal{K}^{\frac{1}{4}}}), \mathcal{E} = \mathcal{O}(\frac{1}{\sqrt{\mathcal{K}}})$, $\Gamma = \mathcal{O}(\frac{1}{\sqrt{\mathcal{K}}})$, we can derive the sublinear convergence rate:

$$\frac{1}{\mathcal{K}} \sum_{k=0}^{\mathcal{K}-1} \mathbb{E}\left[||\nabla\mathcal{R}\left(\eta^{*,k}, \psi^k, \theta^k\right)||^2\right] \leq \mathcal{O}(\frac{1}{\sqrt{\mathcal{K}}}) + \mathcal{O}(\frac{N}{M}) \tag{B.69}$$

**Proof:**

$$\frac{1}{\mathcal{K}} \sum_{k=0}^{\mathcal{K}-1} \mathbb{E}\left[||\nabla\mathcal{R}\left(\eta^{*,k}, \psi^k, \theta^k\right)||^2\right]$$

$$\leq \mathcal{O}(\frac{1}{\sqrt{\mathcal{K}}})[(2\Lambda) + (2L_\psi\sigma_\psi^2 + 2L_\theta\sigma_\theta^2)$$

$$+ 2(1 + L_\psi\alpha_M)H_\psi^2 C + 3(1 + L_\theta\alpha_M)Q_\theta^2 H_\theta^2 C + \frac{3H_\theta^2 C\pi(M,N)}{z^2}(2(1 + L_\psi\alpha_M)L_{\psi\eta}^2 + 3(1 + L_\theta\alpha_M)L_{\theta\eta}^2)$$

$$+ (\frac{3\alpha_M(1 + L_\theta\alpha_M)L^2 d^2}{4\alpha_m} + \frac{3\pi(M,N)L^2 d^2 a_M(1 + L_\psi\alpha_M)L_{\psi\eta}^2}{2a_m z^2} + \frac{9\pi(M,N)L^2 d^2 a_M(1 + L_\theta\alpha_M)L_{\theta\eta}^2}{4a_m z^2})]$$

$$+ \frac{3K^2\pi(M,N)}{z^2}\left[2(1 + L_\psi\alpha_M)L_{\psi\eta}^2 + 3(1 + L_\theta\alpha_M)L_{\theta\eta}^2\right]$$

$$= \mathcal{O}(\frac{1}{\sqrt{\mathcal{K}}}) + \mathcal{O}(\frac{N}{M}) \tag{B.70}$$

# C  EXPERIMENT DETAILS

## C.1  DATASET DETAILS

Our experiments were constricted on public datasets MNIST and CIFAR10:

- **MNIST** LeCun et al. (1998): A benchmark dataset for image classification, comprising 60,000 examples for training and 10,000 examples for testing.
- **CIFAR10** Krizhevsky (2009): Another public dataset for image classification that consists of 60,000 images categorized into 10 classes.

To simulate the VFL scenario, we allocated distinct features to each party based on the described methodology in prior works Luo et al. (2021); Qiu et al. (2022); Fu et al. (2022). We partition the last dimension of the features according to the feature proportion of each client. We use masking to ensure that each client receives distinct features.

## C.2  ADVERSARIAL ATTACK

To validate robustness, we employed a suite of adversarial attack methods. FGSM is a fast, non-iterative attack Kurakin et al. (2016); PGD-$r$ iteratively perturbs input data using gradient information to maximize the model's loss Madry et al. (2017); and CW uses a custom loss function to ensure minimal perturbations while achieving misclassification Carlini & Wagner (2017). CERTIFY (CER) generates adversarial perturbations with Gaussian noise Cohen et al. (2019). For black-box attacks, we combined adversarial methods with zeroth-order optimization: FGSM (ZO-FGSM) and PGD (ZO-PGD) Chen et al. (2017). We also considered scenarios involving a third-party adversary, corrupting embeddings using different client selection strategies, including Thompson Sampling with Empirical Maximum Reward (E-TS) Duanyi et al. (2023) and All Corruption Patterns (ALL).

## C.3  HYPERPARAMETERS

For the parameter updates of both the server and client models, we have adopted the Adam optimizer with a uniform learning rate of $\alpha_\psi = \alpha_\theta = 0.0001$.

Moreover, We follow the hyperparameters choices of Carlini & Wagner (2017); Croce & Hein (2020); Kurakin et al. (2016); Shafahi et al. (2019); Zhang et al. (2019); Zhu et al. (2019) for training.

Table 8: Hyperparameters for Adv. Training

| Dataset | Client Model | batch size | ZOO | | Compress | | Adv. | | DecVFAL | | PGD | FreeAT | FreeLB |
|---------|--------------|-----------|-----|------|----------|-----|------|-------|---------|----|-----|--------|--------|
| | | | $q$ | $\mu$ | type | bit | $\epsilon$ | $\sigma$ | m | n | n | n | n |
| MNIST | MLP | 32 | 100 | 0.05 | scale | 2 | 0.02 | 0.002 | 5 | 10 | 40 | 8 | 40 |
| CIFAR10 | ResNet-18 | 80 | 200 | 0.5 | scale | 2 | 8/255 | 1/255 | 6 | 2 | 10 | 8 | 10 |
| MNIST | ResNet-18 | 32 | 100 | 0.05 | scale | 2 | 0.3 | 0.35 | 6 | 8 | 40 | 8 | 40 |

Table 9: Hyperparameters for Attack

| Dataset | Client Model | ZOO | | FGSM | PGD | | | CW | | | CER | ZO-FGSM | ZO-PGD | | | ALL & E-TS | | |
|---------|--------------|-----|------|------|-----|-------|-------|-----|------|-----|-----|---------|--------|--------|--------|-----|--------|--------|
| | | $q$ | $\mu$ | $\epsilon$ | n | $\epsilon$ | $\sigma$ | n | $\sigma$ | c | $\epsilon$ | $\epsilon$ | n | $\epsilon$ | $\sigma$ | n | $\epsilon$ | $\sigma$ |
| MNIST | MLP | 100 | 0.05 | 16/255 | 40 | 24/255 | 4/255 | 100 | 0.32 | 0.5 | 128/255 | 64/255 | 40 | 96/255 | 12/255 | 10 | 96/255 | 12/255 |
| CIFAR10 | ResNet-18 | 200 | 0.05 | 0.01 | 10 | 10/255 | 1/2550 | 100 | 128/255 | 0.8 | 64/255 | 32/255 | 40 | 32/255 | 2/255 | 1 | 32/255 | 64/255 |
| MNIST | ResNet-18 | 100 | 0.05 | 96/255 | 40 | 64/255 | 2/255 | 100 | 0.8 | 0.5 | 204/255 | 64/255 | 40 | 153/255 | 16/255 | 40 | 128/255 | 16/255 |

## C.4  ENVIRONMENT

In our experiments, we utilized the following software environment: PyTorch version 2.2.1, CUDA version 12.1, and Python version 3.11. The hardware specifications are detailed in Table 10.

Table 10: Hardware Specifications

| Experiment Description | CPU | GPU |
|---|---|---|
| MNIST Robust Training | AMD EPYC 7551P | A4000*1 |
| CIFAR-10 Robust Training | AMD EPYC 7452 | 4090*4 |
| Performance across various NN architectures | Intel E5-2683 v4 | 4090*1 |
| Impact of split position | AMD EPYC 7J13 | 4090*4 |
| Impact of the number of modules | AMD EPYC 7J13 | 4090*4 |
| Impact of the number of the clients | Intel Platinum 8336C | 4090*8 |
| Limitation of the setting of M and N, M = 5 | AMD EPYC 7J13 | 4090*4 |
| Limitation of the setting of M and N, M = 10 | Intel Fold 6430 | 4090*8 |

## C.5 PERFORMANCE ACROSS VARIOUS NN ARCHITECTURES.

We expanded our experiments by incorporating ResNet18 on the MNIST dataset, introducing a different architectural context for evaluating our framework. Similar as experiments in CIFAR-10, the entire model is partitioned into three modules: the first layers of the client models, the remaining layers of the client models, and the server's single-layer perceptron. As shown in Table 11, DecVFAL achieves the best robust performance while requiring only one-seventh of the training time per epoch for PGD adversarial training.

Table 11: Results of MNIST Robust Training with Resnet-18

| Training Methods | Clean Accuracy | White-Box Adv. Atk | | | Black-Box Adv. Atk | | | Third Adv. | | Train Time |
|---|---|---|---|---|---|---|---|---|---|---|
| | | FGSM | PGD | CW | CER | ZO-FGSM | ZO-PGD | ALL | E-TS | (s/epoch) |
| None | 98.66 | 56.56 | 12.26 | 20.99 | 68.37 | 48.67 | 71.43 | 47.96 | 75.71 | **90.86** |
| PGD | 98.23 | 84.73 | 74.74 | 21.16 | 83.98 | 38.21 | 83.69 | 44.56 | 72.49 | 1180.74 |
| FreeAT | 98.44 | 79.47 | 82.20 | **52.33** | 90.82 | **94.84** | 89.04 | 46.52 | 70.15 | 332.63 |
| FreeLB | 98.82 | 70.81 | 40.68 | 31.53 | 80.05 | 36.51 | 81.27 | 65.91 | 87.07 | 1419.10 |
| YOPO | 98.72 | 83.11 | 82.13 | 21.30 | 87.44 | 54.13 | 87.33 | **71.05** | **88.98** | 240.62 |
| DP | 98.63 | 80.20 | 66.63 | 29.80 | 80.63 | 42.68 | 81.55 | 44.77 | 70.31 | 1175.02 |
| MP | 98.12 | 81.38 | 74.47 | 36.31 | 86.42 | 53.64 | 84.84 | 50.38 | 74.10 | 1181.69 |
| Asy-PGD | 98.05 | 79.42 | 76.27 | 27.57 | 85.93 | 42.63 | 84.71 | 57.95 | 80.16 | 1167.09 |
| DecVFAL | **98.98** | **89.00** | **83.20** | 50.80 | **93.91** | 90.95 | **91.17** | 60.22 | 84.14 | **167.89** |

## C.6 EVALUATION UNDER ATTACKS INVOLVING CORRUPTION PATTERN SELECTION

To further assess our framework's resilience in more complex attack scenarios, we conducted experiments on the MNIST dataset using seven clients. Specifically, we evaluated DecVFAL and baseline methods against attacks involving corruption pattern selection. In this setup, adversaries could selectively corrupt client data or communications. The server model remained a single-layer perception. We implemented various corruption patterns, including E-TS, RC, and FC. As shown in Table 12, the results demonstrated that even under these challenging conditions, DecVFAL maintained superior performance compared to baseline methods.

Table 12: Results of evaluation under attacks with various corruption patterns

| Training Methods | White-Box Adv. Atk | | | Black-Box Adv. Atk | | | Third Adversary Atk | | |
|---|---|---|---|---|---|---|---|---|---|
| | Corrupted clients: 1/7 | | | | | | | | |
| | PGD | FGSM | CW | CER | ZO-FGSM | ZO-PGD | E-TS | FC | RC |
| PGD | 92.238 | 94.01 | 93.85 | 94.091 | 94.03 | 93.399 | 88.842 | 88.922 | 88.982 |
| DecVFAL | 95.613 | 96.575 | 96.795 | 96.605 | 96.585 | 96.044 | 93.048 | 93.87 | 93.219 |
| | Corrupted clients: 3/7 | | | | | | | | |
| Training Methods | White-Box Adv. Atk | | | Black-Box Adv. Atk | | | Third Adversary Atk | | |
| | PGD | FGSM | CW | CER | ZO-FGSM | ZO-PGD | E-TS | FC | RC |
| PGD | 79.888 | 87.099 | 93.359 | 94.101 | 92.819 | 92.758 | 77.364 | 78.105 | 77.754 |
| DecVFAL | 86.569 | 92.949 | 96.044 | 96.404 | 95.543 | 94.922 | 84.816 | 85.577 | 84.685 |
| | Corrupted clients: 5/7 | | | | | | | | |
| Training Methods | White-Box Adv. Atk | | | Black-Box Adv. Atk | | | Third Adversary Atk | | |
| | PGD | FGSM | CW | CER | ZO-FGSM | ZO-PGD | E-TS | FC | RC |
| PGD | 64.724 | 80.689 | 91.526 | 93.279 | 90.935 | 91.587 | 69.03 | 68.53 | 69.111 |
| DecVFAL | 78.235 | 87.31 | 91.987 | 96.044 | 93.049 | 94.121 | 75.972 | 76.062 | 76.322 |

