# OpenReview forum: "Accelerate Vertical Federated Adversarial Learning with  Dual-level Decoupled  Backpropagation"
_ICLR.cc/2025/Conference — Submitted to ICLR 2025_

### Official Review · Reviewer_zWdE · 2024-11-01

**Soundness:** 2
**Presentation:** 1
**Contribution:** 2
**Rating:** 3
**Confidence:** 2

**Summary:**

The paper proposes to use decoupled forward and backward propagation in vertical federated learning with adversarial attacks. The main idea is to use cached variables to circumvent the lock in back-propagation, thus updating the adversarial samples with parallel computation. The proposed DecVFAL demonstrates improved performance on MNIST and CIFAR10.

**Strengths:**

The pictorial presentation is elegant. Numerical simulations show promising result in terms of robustness to attacks and running time.

**Weaknesses:**

- The combination of VFL with decoupled backward propagation does not seem super exciting. The decoupled neural inference is well-established in Jaderberg et al. and follow-up works. The VFL framework used in the paper follows Wang et al. Combining them does not lead to speed-ups specific to VFL. Authors should clarify their contributions.

- The writing can be improved. The meanings of most notations are not clear until Table 6 in the appendix, which can easily result in confusion.

- The definition of $p^{m,N}_{\mathcal{M}_1,i}$ in (4.2) is unclear even after reading the appendix.

- Theorem 1 does not generate clear insight: authors should clarify how the convergence result shows running time is improved by decoupled backward propagation.


Minor:

- Inline citation should use citep instead of cite.

**Questions:**

- In general, the verbal part of the paper is uneasy to follow. Authors should consider simplifying the notation and highlighting the contribution in simpler language.

- In Table 1, it is surprising to see that DecVFAL generates better performance than PGD. Since DecVFAL uses approximate adversarial examples, one would expect the performance to be worse than PGD, which uses optimized adversarial examples. Can authors provide more intuitions?

---

### Official Review · Reviewer_do28 · 2024-11-01

**Soundness:** 3
**Presentation:** 2
**Contribution:** 2
**Rating:** 5
**Confidence:** 3

**Summary:**

This paper proposes DecVFAL, which is an accelerated solver for vertical federated adversarial learning (VFAL) problems. DecVFAL features a lazy sequential backpropagation technique and a decoupled parallel backpropagation scheme to reduce the computation cost and improve the parallelizability over standard VFAL solvers.

**Strengths:**

The paper is in general well-written. The convergence properties of the proposed algorithm are analyzed under standard assumptions. The numerical results suggest that the proposed algorithm outperforms existing approaches on simple benchmarks.

**Weaknesses:**

1. The VFAL problem considered in this paper seems to me a rather niche problem, which considers scenarios that multiple clients hold different features of the exact same data samples and the server is able to evaluate the loss (e.g., has access to the label) for all data samples. It is not very clear to me what impact an improved VFAL solver can make. It would be helpful if the authors could provide concrete realistic use cases and demonstrate the benefits of the proposed algorithm on relevant toy problems.

2. Although pseudocodes and figures are provided to help reader understand the proposed algorithm, the notation is sometimes confusing. For example, in the last paragraph of p.4, the number of modules seems to me should be $K$ rather than ${\mathcal{M}_K}$ and the counter $k$ should go from 1 to $K$ instead of $\mathcal{M}_k$ in line 6, Algorithm 1. Variables should be properly initialized in the pseudocode. Also, the key objects in the theoretical results, e.g., the adversarial loss $\mathcal{A}$, $\mathcal{R}$, are defined only in the appendix, which makes it difficult to interpret the theoretical results and assess their values. Please fix the notation and consider to include necessary definitions for interpreting theoretical results.

3. The paper does not address the communication cost of the proposed algorithm, which is one of the primary bottlenecks of general federated learning problems. I understand that the total communication is reduced if the algorithm can converge in fewer epochs when the amount of information transferred in each communication round is comparable to existing methods, but adding a remark to discuss the communication cost would be helpful.

4. The experiments consider VFAL for MNIST and CIFAR10 with a rather homogeneous splitting of images between clients. It would be more interesting and perhaps more realistic to test benchmarks with multimodal data in which each client holds a mode of the data samples. The numbers of clients considered in the experiments (3, 5, 7) are small.

**Questions:**

1. I do not see a clear connection of the proposed algorithm to federated learning. Specifically, the algorithm seems to be designed to address issues that occur in layer-wise model parallelism (when a network model is distributed layer-wisely to multiple devices/machines), which could happen in a non-federated scenario with no client/server separation. Is there something in the algorithm that is specific to the multiple client case that I missed? If so, it would be helpful to clearly identify them in the paper and discuss how these features address challenges specific to federated learning scenarios, as opposed to general distributed learning with model parallelism.

2. Why is there no multiple clients in Figure 1? If it makes sense, I suggest including a version that shows multiple clients to better illustrate the federated learning aspect of this approach.

**Details Of Ethics Concerns:**

None.

---

### Official Review · Reviewer_9eNW · 2024-11-04

**Soundness:** 2
**Presentation:** 3
**Contribution:** 2
**Rating:** 5
**Confidence:** 3

**Summary:**

This paper proposes DecVFAL, a framework that addresses the Vertical Federated Adversarial Learning (VFAL) problem. Specifically, DecVFAL employs a dual-level decoupled mechanism to enable lazy sequential and decoupled parallel backpropagation, which accelerates the VFAL process.
Overall, the paper is relatively easy to follow, and the theoretical analysis is quite interesting. However, it needs to be validated on larger datasets and in more realistic settings.

**Strengths:**

1. The paper is generally well-written and relatively easy to follow.

2. Rigorous theoretical analysis demonstrates that DecVFAL achieves a fast convergence rate.

**Weaknesses:**

1. The validation was only conducted on MNIST and CIFAR-10; it is necessary to test the effectiveness of DecVFAL on larger datasets.
2. Can DecVFAL be validated in more realistic client settings, such as scenarios with 50 or 100 clients?
3. It is better to dedicate more space to discussing the differences between VFAL and FAL, as well as the more complex challenges faced by VFAL.

**Questions:**

1. Why is adversarial training needed in Vertical Federated Learning? Is there a difference between Vertical Federated Adversarial Learning (VFAL) and Federated Adversarial Learning (FAL)? What are the different challenges facing VFAL compared to FAL?
2. Are there any real-world application scenarios for VFAL? What are the applications?
3. Is the trained classifier maintained on the server? How is inference or prediction conducted using the trained classifier?
4. In the dual-level decoupled mechanism introduced by DecVFAL, what do "dual-level" and "decoupled" mean?
5. The abbreviation "PGD" appears for the first time on page 2, but its full name is not introduced.
6. Some equations lack punctuation at the end, such as Eqs. (4.1), (4.3), (4.4), (5.1), (5.2).
7. In Lemma 1, equation 4.2 should end with a comma, and "Where" should be lowercase.
8. Section 4.3 emphasizes the importance of "lazy sequential backpropagation" for DecVFAL. It would be better to add a comparison in the experiment between DecVFAL with and without the "lazy sequential backpropagation" module to highlight its advantages in efficiency.
9. “Under Assumptions (1, 2, 3, 4)” —> “Under Assumptions 1-4”
10. Can DecVFAL's effectiveness be validated on larger or more complex datasets, such as CIFAR-100?
11. Is DecVFAL still effective with more clients, like 50 or 100 clients?
12. Does the convergence rate of Vertical Federated Learning differ from that of original Federated Learning?
13. In Vertical Federated Learning, does the sample size among different clients affect convergence rate? Suppose there are 100 clients, with one client holding 90% of the total samples and the other clients holding only 10%. Would the convergence rate slow down in this case?
14. It is recommended to add some explanations for "VFL with PGD (left)" in the label of Figure 1, and briefly indicate how it differs from "DecVFAL (right)."

**Details Of Ethics Concerns:**

N.A.

---

### Meta-Review · Area_Chair_87xH · 2024-12-13

**Metareview:**

The authors did not respond to the reviewers. Based on the reviews and recommendations, the paper gets rejected.

**Additional Comments On Reviewer Discussion:**

NA

---

### Decision · Program_Chairs · 2025-01-22

Reject